# Genomic analysis of two phlebotomine sand fly vectors of *Leishmania* from the New and Old World

**Frédéric Labbé**[1¤], **Maha Abdeladhim**[2], **Jenica Abrudan**[3], **Alejandra Saori Araki**[4], **Ricardo N. Araujo**[5], **Peter Arensburger**[6], **Joshua B. Benoit**[7], **Reginaldo Pecanha Brazil**[8], **Rafaela V. Bruno**[4], **Gustavo Bueno da Silva Rivas**[4,9], **Vinicius Carvalho de Abreu**[10], **Jason Charamis**[11,12], **Iliano V. Coutinho-Abreu**[13], **Samara G. da Costa-Latgé**[4], **Alistair Darby**[14], **Viv M. Dillon**[14], **Scott J. Emrich**[15], **Daniela Fernandez-Medina**[16], **Nelder Figueiredo Gontijo**[5], **Catherine M. Flanley**[1], **Derek Gatherer**[17], **Fernando A. Genta**[4], **Sandra Gesing**[18], **Gloria I. Giraldo-Calderón**[1,19], **Bruno Gomes**[4], **Eric Roberto Guimaraes Rocha Aguiar**[10], **James G. C. Hamilton**[17], **Omar Hamarsheh**[20], **Mallory Hawksworth**[1], **Jacob M. Hendershot**[7], **Paul V. Hickner**[21], **Jean-Luc Imler**[22], **Panagiotis Ioannidis**[12], **Emily C. Jennings**[7], **Shaden Kamhawi**[2], **Charikleia Karageorgiou**[12,23], **Ryan C. Kennedy**[1], **Andreas Krueger**[24,25], **José M. Latorre-Estivalis**[26], **Petros Ligoxygakis**[27], **Antonio Carlos A. Meireles-Filho**[4], **Patrick Minx**[28], **Jose Carlos Miranda**[29], **Michael J. Montague**[30], **Ronald J. Nowling**[31], **Fabiano Oliveira**[2], **João Ortigão-Farias**[32], **Marcio G. Pavan**[4,33], **Marcos Horacio Pereira**[5], **Andre Nobrega Pitaluga**[34], **Roenick Proveti Olmo**[10], **Marcelo Ramalho-Ortigao**[35], **José M. C. Ribeiro**[2], **Andrew J. Rosendale**[9], **Mauricio R. V. Sant'Anna**[5], **Steven E. Scherer**[36], **Nágila F. C. Secundino**[37], **Douglas A. Shoue**[1], **Caroline da Silva Moraes**[4], **João Silveira Moledo Gesto**[4], **Nataly Araujo Souza**[38], **Zainulabeuddin Syed**[39], **Samuel Tadros**[1], **Rayane Teles-de-Freitas**[4], **Erich L. Telleria**[31,40], **Chad Tomlinson**[41], **Yara M. Traub-Csekö**[32], **João Trindade Marques**[9], **Zhijian Tu**[42], **Maria F. Unger**[43], **Jesus Valenzuela**[2], **Flávia V. Ferreira**[44], **Karla P. V. de Oliveira**[10], **Felipe M. Vigoder**[45], **John Vontas**[12,46], **Lihui Wang**[28], **Gareth D. Weedall**[47,48], **Elyes Zhioua**[49], **Stephen Richards**[36], **Wesley C. Warren**[50], **Robert M. Waterhouse**[51], **Rod J. Dillon**[17], **Mary Ann McDowell**[1]*

1 Eck Institute for Global Health, Department of Biological Sciences, University of Notre dame, Notre Dame, Indiana, United States of America, 2 Vector Molecular Biology Section, Laboratory of Malaria and Vector Research, National Institute of Allergy and Infectious Diseases, National Institutes of Health, Rockville, Maryland, United States of America, 3 Genomic Sciences & Precision Medicine Center (GSPMC), Medical College of Wisconsin, Milwaukee, Wisconsin, United States of America, 4 Laboratório de Bioquímica e Fisiologia de Insetos, IOC, FIOCRUZ, Rio de Janeiro, Brazil, 5 Laboratório de Fisiologia de Insetos Hematófagos, Universidade Federal de Minas Gerais, Instituto de Ciencias Biológicas, Departamento de Parasitologia, Pampulha, Belo Horizonte, Brazil, 6 Department of Biological Sciences, California State Polytechnic University, Pomona, California, United States of America, 7 Department of Biological Sciences, University of Cincinnati, Cincinnati, Ohio, United States of America, 8 Laboratório de Doenças Parasitárias, Instituto Oswaldo Cruz, Rio de Janeiro, Brazil, 9 Department of Biology and Center for Biological Clocks Research, Texas A&M University, College Station, Texas, United States of America, 10 Department of Biochemistry and Immunology, Instituto de Ciências Biológicas, Universidade Federal de Minas Gerais, Belo Horizonte, Brazil, 11 Department of Biology, University of Crete, Voutes University Campus, Heraklion, Greece, 12 Molecular Entomology Lab, Institute of Molecular Biology and Biotechnology, Foundation for Research and Technology Hellas (FORTH), Heraklion, Greece, 13 Division of Biological Sciences, Section of Cell and Developmental Biology, University of California, San Diego, California, United States of America, 14 Institute of Integrative Biology, The University of Liverpool, Liverpool, United Kingdom, 15 Department of Electrical Engineering and Computer Science, University of Tennessee, Knoxville, Tennessee, United States of America, 16 School of Applied Mathematics, Getulio Vargas Foundation, Rio de Janeiro, Brazil, 17 Division of Biomedical & Life Sciences, Faculty of Health & Medicine, Lancaster University, Lancaster, United Kingdom, 18 Discovery Partners Institute, University of Illinois Chicago, Chicago, Illinois, United States of America, 19 Dept. Ciencias Biológicas & Dept. Ciencias Básicas Médicas, Universidad Icesi, Cali, Colombia, 20 Department of Life Sciences, Faculty of Science and Technology, Al-Quds University, Jerusalem, Palestine, 21 USDA-ARS Knipling-Bushland U.S. Livestock Insects Research Laboratory and Veterinary Pest Genomics Center, Kerrville, Texas, United States of America, 22 CNRS-UPR9022 Institut de



**Data Availability Statement:** Sequenced reads sequence reads were deposited in the NCBI SRA under Bioproject accession number PRJNA20279 for Lutzomyia longipalpis and PRJNA20293 for

Phlebotomus papatasi. The GeneBank Ph. papatasi Ppap_1.0 assembly accession number is GCA_000262795.1 and the Lu. longipalpis Llon_1.0 assembly accession number is GCA_000265325.1. Both genome assemblies and associated annotations are hosted on VectorBase.

**Funding:** The work was supported by two grants from the National Human Genome Research Institutes U54-HG003079 to WCW and U54-HG003273 to SR. The funders had no role in study design, data collection and analysis, decision to publish, or preparation of the manuscript.

**Competing interests:** The authors have declared that no competing interests exist.

Biologie Moléculaire et Cellulaire and Faculté des Sciences de la Vie-Université de Strasbourg, Strasbourg, France, **23** Genomics Group – Bioinformatics and Evolutionary Biology Lab, Department of Genetics and Microbiology, Autonomous University of Barcelona, Barcelona, Spain, **24** Medical Entomology Branch, Dept. Microbiology, Bundeswehr Hospital, Hamburg, Germany, **25** Medical Zoology Branch, Dept. Microbiology, Central Bundeswehr Hospital, Koblenz, Germany, **26** Laboratorio de Insectos Sociales, Instituto de Fisiología, Biología Molecular y Neurociencias, Universidad de Buenos Aires - CONICET, Buenos Aires, Argentina, **27** Laboratory of Cell Biology, Development and Genetics, Department of Biochemistry, University of Oxford, Oxford, United Kingdom, **28** Donald Danforth Plant Science Center, Olivette, Missouri, United States of America, **29** Laboratório de Imunoparasitologia, CPqGM, Fundação Oswaldo Cruz, Bahia, Brazil, **30** Department of Neuroscience, Perelman School of Medicine, University of Pennsylvania, Philadelphia, Pennsylvania, United States of America, **31** Department of Electrical Engineering and Computer Science, Milwaukee School of Engineering, Milwaukee, Wisconsin, United States of America, **32** Instituto Oswaldo Cruz – Fiocruz, Rio de Janeiro, Brazil, **33** Laboratório de Transmissores de Hematozoários, IOC, FIOCRUZ, Rio de Janeiro, Brazil, **34** Laboratório de Biologia Molecular de Parasitas e Vetores, Instituto Oswaldo Cruz/FIOCRUZ, Rio de Janeiro, Brazil, **35** F. Edward Hebert School of Medicine, Department of Preventive Medicine and Biostatistics, Uniformed Services University of the Health Sciences (USUHS), Bethesda, Maryland, United States of America, **36** Human Genome Sequencing Center, Baylor College of Medicine, Houston, Texas, United States of America, **37** Laboratory of Medical Entomology, René Rachou Institute-FIOCRUZ, Belo Horizonte, Brazil, **38** Laboratory Interdisciplinar em Vigilancia Entomologia em Diptera e Hemiptera, Fiocruz, Rio de Janeiro, Brazil, **39** Department of Entomology, University of Kentucky, Lexington, Kentucky, United States of America, **40** Department of Parasitology, Faculty of Science, Charles University, Prague, Czech Republic, **41** McDonnell Genome Institute, Washington University School of Medicine, St. Louis, Missouri, United States of America, **42** Fralin Life Science Institute and Department of Biochemistry, Virginia Tech, Blacksburg, Virginia, United States of America, **43** Department of Chemical and Biomolecular Engineering, University of Notre Dame, Notre Dame, Indiana, United States of America, **44** Department of Microbiology, Instituto de Ciências Biológicas, Universidade Federal de Minas Gerais, Belo Horizonte, Brazil, **45** Universidade Federal do Rio de Janeiro, Instituto de Biologia. Rio de Janeiro, Brazil, **46** Pesticide Science Lab, Department of Crop Science, Agricultural University of Athens, Athens Greece, **47** Vector Biology Department, Liverpool School of Tropical Medicine (LSTM), Liverpool, United Kingdom, **48** School of Biological and Environmental Sciences, Liverpool John Moores University, Liverpool, United Kingdom, **49** Vector Ecology Unit, Institut Pasteur de Tunis, Tunis, Tunisia, **50** Department of Animal Sciences, Department of Surgery, Institute for Data Science and Informatics, University of Missouri, Columbia, Missouri, United States of America, **51** Department of Ecology & Evolution and Swiss Institute of Bioinformatics, University of Lausanne, Lausanne, Switzerland

¤ Current Address: CIRAD, UMR PVBMT, Saint Pierre, France
* mcdowell.11@nd.edu

## Abstract

Phlebotomine sand flies are of global significance as important vectors of human disease, transmitting bacterial, viral, and protozoan pathogens, including the kinetoplastid parasites of the genus *Leishmania*, the causative agents of devastating diseases collectively termed leishmaniasis. More than 40 pathogenic *Leishmania* species are transmitted to humans by approximately 35 sand fly species in 98 countries with hundreds of millions of people at risk around the world. No approved efficacious vaccine exists for leishmaniasis and available therapeutic drugs are either toxic and/or expensive, or the parasites are becoming resistant to the more recently developed drugs. Therefore, sand fly and/or reservoir control are currently the most effective strategies to break transmission. To better understand the biology of sand flies, including the mechanisms involved in their vectorial capacity, insecticide resistance, and population structures we sequenced the genomes of two geographically widespread and important sand fly vector species: *Phlebotomus papatasi*, a vector of *Leishmania* parasites that cause cutaneous leishmaniasis, (distributed in Europe, the Middle East and North Africa) and *Lutzomyia longipalpis*, a vector of *Leishmania* parasites that cause visceral leishmaniasis (distributed across Central and South America). We categorized and

curated genes involved in processes important to their roles as disease vectors, including chemosensation, blood feeding, circadian rhythm, immunity, and detoxification, as well as mobile genetic elements. We also defined gene orthology and observed micro-synteny among the genomes. Finally, we present the genetic diversity and population structure of these species in their respective geographical areas. These genomes will be a foundation on which to base future efforts to prevent vector-borne transmission of *Leishmania* parasites.

## Author summary

The leishmaniases are a group of neglected tropical diseases caused by protist parasites from the Genus *Leishmania*. Different *Leishmania* species present a wide clinical profile, ranging from mild, often self-resolving cutaneous lesions that can lead to protective immunity, to severe metastatic mucosal disease, to visceral disease that is ultimately fatal. *Leishmania* parasites are transmitted by the bites of sand flies, and as no approved human vaccine exists, available drugs are toxic and/or expensive and parasite resistance to them is emerging, new dual control strategies to combat these diseases must be developed, combining interventions on human infections and integrated sand fly population management. Effective vector control requires a comprehensive understanding of the biology of sand flies. To this end, we sequenced and annotated the genomes of two sand fly species that are important leishmaniasis vectors from the Old and New Worlds. These genomes allow us to better understand, at the genetic level, processes important in the vector biology of these species, such as finding hosts, blood-feeding, immunity, and detoxification. These genomic resources highlight the driving forces of evolution of two major *Leishmania* vectors and provide foundations for future research on how to better prevent leishmaniasis by control of the sand fly vectors.

## Introduction

Phlebotomine sand flies are a group of blood-feeding Diptera that vary widely in their geographic distribution, ecology, and the pathogens they transmit. They serve as vectors for several established, emerging, and re-emerging infectious diseases, transmitting protist, bacterial and viral pathogens. The most important of the sand fly transmitted pathogens belong to the genus *Leishmania* which cause a spectrum of disease in humans known as leishmaniasis, that account for an estimated 2.4 million disability-adjusted life-years (DALYs) [1] and 40,000 deaths annually [2]. These statistics are likely to be underestimated due to misdiagnosis, underreporting, and lack of surveillance systems in many of the affected countries. Political instability, urbanization, and climate change are expanding *Leishmania*-endemic regions and increasing the risk of epidemics world-wide [3]. These factors coupled with the increase of visceral disease and HIV co-infection, have led the World Health Organization to classify leishmaniasis as one of the world's epidemic-prone diseases [4].

Leishmaniasis occurs worldwide, in 98 countries over five continents, with 310 million people at risk of contracting the infection [2]. Leishmaniasis is a collective term for a group of distinct clinical manifestations ranging from mild, often self-resolving cutaneous lesions that can lead to protective immunity, to disseminated lesions that do not heal spontaneously, to destruction of the mucous membranes of the nose, mouth, and pharynx, to life-threatening

visceral disease. The clinical profile depends on a variety of factors, including vector biology, host immunity, and parasite characteristics; with the *Leishmania* species that causes the infection being the primary determinant. The two primary clinical forms are cutaneous leishmaniasis (CL) and visceral leishmaniasis (VL). The primary *Leishmania* species that cause CL are *Leishmania major*, *Leishmania infantum*, *Leishmania tropica*, and *Leishmania aethipica* in the Old World and *Leishmania amazonensis*, *Leishmania braziliensis*, *Leishmania guyanensis*, *Le. infantum*, *Leishmania mexicana*, and *Leishmania panamensis* in the New World. VL is primarily caused by *Leishmania donovani* in Asia and Africa and *Le. infantum* in the Middle East, central Asia, South and Central America, and the Mediterranean Basin.

There are approximately 35 proven, and an additional 63 suspected, vectors of at least 40 different *Leishmania* species to humans [5,6]. *Phlebotomus* species are the primary *Leishmania* vectors in the Old World and *Lutzomyia* species are responsible for transmitting leishmaniasis throughout the Americas [7]. There is a close ecological association, if not co-evolutionary relationship [8,9], between *Leishmania* species and their specific vectors such that generally a single sand fly species transmits a single *Leishmania* species under natural conditions. Some sand flies, however, can transmit a range of *Leishmania* species under experimental conditions [10]. This difference has given rise to the concept of "restricted" and "permissive" vectors [11]. For example, *Phlebotomus papatasi* is a restrictive vector, transmitting only *Le. major* parasites [12]. *Lutzomyia longipalpis* (*s.l.*) is considered a permissive vector in laboratory conditions, but only transmits *Le. infantum* naturally [12].

These vectors are part of the Diptera which is an extremely species-rich and ecologically diverse order of insects and contains the vectors of many of the most important pathogens of man and his domesticated animals. Both phlebotomine sand flies (family Psychodidae) and mosquitoes (Culicidae) are specified as members of distinct infra-orders within the suborder Nematocera. While the Nematocera grouping is paraphyletic, the relationships between infra-orders remains to be elucidated [13]. Some studies generated topologies with Psychodomorpha (sand flies) and Culicomorpha (mosquitoes and black flies) as sister groups [14], whereas, others place sand flies nearer to the muscoid flies (Ephydroidea) [15]. The internal relationships within the assemblage that includes Psychodidae also remains a matter of debate [16].

It is postulated that the close evolutionary relationship between sand fly species and the *Leishmania* species that they transmit may have epidemiological implications for leishmaniasis [17]. For example, there are three primary zymodemes of *Le. major* that have limited geographical distributions such that the prevalent zymodeme in a particular area overlaps with the distribution of one primary population of *Ph. papatasi* [18]. *Ph. papatasi* has a wide geographical distribution, ranging from Morocco to the Indian subcontinent and from southern Europe to central and eastern Africa. Given the wide ecological and geographic distribution of *Ph. papatasi* populations [19], coupled with the low dispersal capacity of these sand flies [12], it is likely that there is limited gene flow between populations and significant genetic structuring between populations. While previous studies demonstrated relatively low genetic differentiation between *Ph. papatasi* populations separated by large geographical distances [9,20], more recent studies have identified genetic differentiation between geographically separated populations [18,21–24] and local differentiation [25]. Microsatellite analysis, in particular, revealed two distinct genetic clusters of *Ph. papatasi* (A & B) with further substructure within each population that correlated with geographical origin (A1-5 and B1 &2) [18,23].

While elucidating the drivers leading to reproductive isolation and speciation remains a challenge, there is strong evidence that *Lu. longipalpis* is undergoing incipient speciation in Brazil with various levels of differentiation between siblings of the complex [26]. The Brazilian populations of *Lu. longipalpis* can be divided into three groups based on analysis of their primary copulatory songs which start during mating immediately after the male clasps the female.

The males of one group produce Burst-type mating songs the second, more heterogeneous group, has populations which produce different subtypes of Pulse-type songs. The third group, "mix-type" has characteristics from the other Burst and Pulse types but has sufficient significant differences in all measured characteristics to enable them to be differentiated from the other types [27–29]. Acoustic communication in insects is mostly associated with attraction and/or recognition during courtship, prior to copulation. In *Lu. longipalpis* (*s.l.*), sound production starts when copulation has commenced and contributes to insemination success indicating that it is directly linked to reproductive success [30].

Male *Lu. longipalpis* produce sex-aggregation pheromones, volatile chemicals that attract females to male selected mating sites over long distances [31]. Analysis of structure and quantity of these chemicals indicates that there are at least 5 different pheromone types possibly representing cryptic species of *Lu. longipalpis* in South and Central American countries [32–34] and analysis of molecular correlates [single nucleotide polymorphisms (SNPs) and copy number variation (CNVs)] in the chemosensory genome confirms that these populations have significant genetic differences [35]. The structures of the sex-aggregation pheromones of members of the complex that have been elucidated fall into 2 classes; diterpenes, which have the molecular formula $C_{20}H_{32}$ and molecular weight (mw) 272 gmol$^{-1}$ and methylsesquiterpenes with the molecular formula $C_{16}H_{32}$ and mw 218 gmol$^{-1}$ [32]. One of the diterpenes, has been characterized as sobralene (SOB) [36] and two of the methylsesquiterpenes as 3-methyl-$\alpha$-himachalene (3M$\alpha$H) and (*S*)-9-methylgermacrene-B (9MGB). These compounds are found only in populations of *Lu. longipalpis*.

Although the sex-aggregation pheromones of *Lu. longipalpis* (*s.l.*) share a biosynthetic origin the methylsesquiterpenes are derived from a 15-carbon precursor, farnesyl diphosphate and six of the seven enzymes of the mevalonate-pathway, plus enzymes involved in sesquiterpenoid biosynthesis, have been found in 9MGB-producing *Lu. longipalpis* [37] whereas the diterpenes are derived via a 20-carbon precursor, geranylgeranyl diphosphate [38].

Crossing experiments between sympatric and allopatric populations of different members of the *Lu. longipalpis* species complex revealed reproductive isolation due to both pre-mating and copulatory mechanisms [39,40]. Hickner *et al*. 2020 provided genomic insights into the chemoreceptor genome repertoire underlying behavioral evolution of sexual communication in the *Lu. longipalpis* populations, but whole-genome analyses could improve the identification of loci related to critical traits such as vectorial capacity, host preference, and insecticide resistance [35].

Despite the potential importance for influencing *Leishmania* development and survival in the gut, the sand fly immune response is poorly studied. To date, work has been largely restricted to the study of defensins [41–44]. However, gene depletion via RNAi of the negative regulator of the Immune Deficiency (IMD) pathway caspar [45] led to a reduction in *Leishmania* population in the gut of *Lu. longipalpis*. While the knockout of relish, the transcription factor of the IMD pathway, resulted in the increase of *Leishmania* and bacteria in *Ph. papatasi* [46].

Adaptation to hematophagy presents many challenges to insects, including avoiding the physiological responses of the host that interfere with obtaining a blood meal, digestion of the blood, and excretion of the excess water contained in the blood meal. Sand flies have evolved a complex cocktail of pharmacologically active salivary molecules to facilitate blood feeding that have been extensively characterized [47].

Many important aspects in sand fly biology such as hematophagy and host seeking are controlled by the biological clock [48]. In *Lu. longipalpis*, the main clock genes and their expression pattern throughout the day have been previously characterized [49,50]. However, the molecular regulation of circadian rhythms is poorly understood in sand flies. Yuan

*et al.* 2007 proposed three clock models based on the presence of the cryptochrome (CRY) proteins, CRY1 and CRY2 [51]. In the *Drosophila* clock model, only CRY1, which acts as a blue-light photoreceptor [52], is present. In the butterfly model, CRY1 also acts as a photo-receptor and CRY2, which is a mammalian–like transcriptional repression, dimers with PER to repress CLK/CYC activity. In the bee model, there is only CRY2, which seems to act as a repressor together with PER and some other molecule that is not CRY1 that acts as photoreceptor.

A central inquiry of evolutionary biology is elucidating drivers of speciation, however, defining species boundaries and identifying the genetic architecture that leads to reproductive isolation has been a challenge. Understanding of the mechanisms of vectorial capacity, adaptation to changing ecological environments, and insecticide resistance has epidemiological consequences for the integrated management of sand fly populations that is the cornerstone of leishmaniasis control [53]. To begin to explore the driving forces of evolution of two important phlebotomine sand fly vectors from the *Psychodidae* family (*Phlebotominae* subfamily), *Ph. papatasi* and *Lu. longipalpis* (*s.l.*), that exhibit distinct distributions, behavior, and pathogen specificity, we sequenced and analyzed their whole-genomes using comparative genomics approaches. We manually curated a number of gene families with key roles in processes such as immunity, blood-feeding, chemosensation, detoxification, and circadian biology to provide a basis for studying and understanding sand flies as *Leishmania* vectors. Moreover, as a better understanding of the population structure of geographically separated vector populations is necessary, we also assessed the population structure of *Ph. papatasi* and *Lu. Longipalpis* by collecting and sequencing individual field-collected specimens sampled over a large geographical range in the Middle East and North Africa, and Brazil, respectively. Our results provide significant advances in our understanding of the genetics underlying the population structure and provide a foundation for future molecular comparative studies of these two medically important vectors.

## Methods

### Ethics statement

The study protocol was approved by the Institutional Animal Care and Use Committee at the University of Notre Dame (#07–052).

### Laboratory colonies

**Phlebotomus papatasi.** To avoid confounding effects due to genetic polymorphisms, we used a colony of *Ph. papatasi* (Israeli strain) for the genome assembly. This colony was originally established in the 1970s and given to Walter Reed Army Institute of Research (WRAIR) in 1983 from the Hebrew University, Jerusalem and transferred to the University of Notre Dame in 2006. At several times since establishment in the laboratory, the colony has fluctuated in population size and has been expanded from a relatively low number of files, therefore, this colony may have reduced heterozygosity. Sand flies were reared by the method of Modi and Tesh [54].

**Lutzomyia longipalpis.** *Lu. longipalpis* Jacobina strain was used for the genome assembly. This colony was originally established at the Liverpool School of Tropical Medicine by Richard Ward in 1988 from flies caught in Jacobina, Bahia State, Brazil. This colony also was expanded from a small number of flies several times since establishment. Flies were reared under standardized laboratory conditions [54], *i.e.* under controlled temperature (27 ± 2°C), humidity (>80%), and photoperiod (8 hours light/16 hours darkness) [54].

### Field collections

**Phlebotomus papatasi.** *Ph. papatasi* were collected from three different locations: Tunisia, Egypt, and Afghanistan. In Tunisia, samples were collected in 2013 from the village of Felta located in an arid biogeographical area in Central Tunisia (35˚16'N, 9˚26'E). In North Sinai Egypt, samples were collected in Om Shikhan (30˚50'N, 34˚10'E), located approximately 340 km east of Cairo, 80 km inland from the Mediterranean coast, and 30 km west of the Israeli border in 2007. In Afghanistan, samples were collected in 2010 in and around a German military camp located near the airport of Mazar-e Sharif (36˚43'N, 67˚14'E). This site is located at 400m altitude north of the Hindukush Mountains and approximately 50km south of the Uzbekistan border. Sand flies were trapped using CDC-style light traps between 17:00 and 07:00.

**Lutzomyia longipalpis.** *Lu. longipalpis* were collected in 2014 from six different locations in Brazil (Fig 1). Samples were collected from three allopatric populations: Jacobina, Bahia State ($11^0$10'S $40^0$31'W), (3MαH), Lapinha Cave, Minas Gerais State ($19^0$38'S $43^0$53'W) (9MGB), Marajó Island, Pará State (0˚56'S 49˚38'W) (SOB), and two sympatric populations from Sobral, Ceará State ($34^0$41'S $40^0$20'W), denoted as S1S (9MGB) and S2S (SOB).. For comparison of male copulatory courtship songs, flies were also collected from Olindina (11˚ 29' S 38˚ 22' W) and Araci (11˚ 09' S 39˚ 01' W), sites near Jacobina. Sand flies were trapped using CDC-style light traps baited with $CO_2$ between 18:00 and 06:00 and transported to the laboratory. Analyses of male copulatory courtship songs was carried out by as previously described [27]. The recordings were performed by using males and females from laboratory colonies established from wild-collected flies from Lapinha and Sobral and from Araci and Olindina.

### Nucleic acid isolation

Genomic DNA from female sand flies was isolated from pools of flies or from single insects for population genetics analysis. For pooled insects, DNA was suspended in 50 μl of the hydration solution using a Tissue DNA isolation kit (GE HealthCare LifeSciences).

To generate an extensive RNA-seq coverage to allow for quality gene prediction, RNA was obtained from both sexes one-, three-, and ten-days post emergence, during development, and adult females post blood-feeding (6, 24, and 96 hours for *Ph. papatasi* and 6, 24, and 144 hours for *Lu. longipalpis*) on uninfected and *Leishmania* [*Le. major* (MHOM/IL/81/Friedlin) for *Ph. papatasi* and *Le. infantum* (MHOM/BR/76/M4192) for *Lu. longipalpis*] infected mouse blood. Total RNA was extracted using a RNAeasy Mini Kit (Qiagen).

### Genome sequencing and assembly

**Phlebotomus papatasi.** Sequencing and assembly for *Ph. papatasi* were performed by the Genome Institute, Washington University School of Medicine. The assembly was built with the–het option, using the Newbler assembler test release 2.6RC02 from an input of ~22.5X total sequence coverage with Sanger and 454 reads including 15.1X of whole-genome shotgun reads, 4.4X 3 kb clone inserts, 3.0X 8 kb inserts and 0.01X BAC-end read pairs. Whole-genome shotgun Illumina paired-end reads (300 bp inserts) were sequenced to 20X coverage for gap closing. The fragment and 3 kb data were generated from a single sand fly after whole-genome amplification, while the 8kb data were derived from multiple flies. The 0.1X of Sanger 3,730 BAC end sequences (28,902 reads) were also derived from multiple flies.

Prior to submission to NCBI, this assembly was screened for contamination as previously described [55] by using MegaBLAST [56] against bacterial and vertebrate genome databases, resulting in the removal of 247 contigs. Heterozygous contigs were removed or merged reducing the assembled genome size from 364 Mb to 345 Mb. A total of 5,661 gaps were closed and

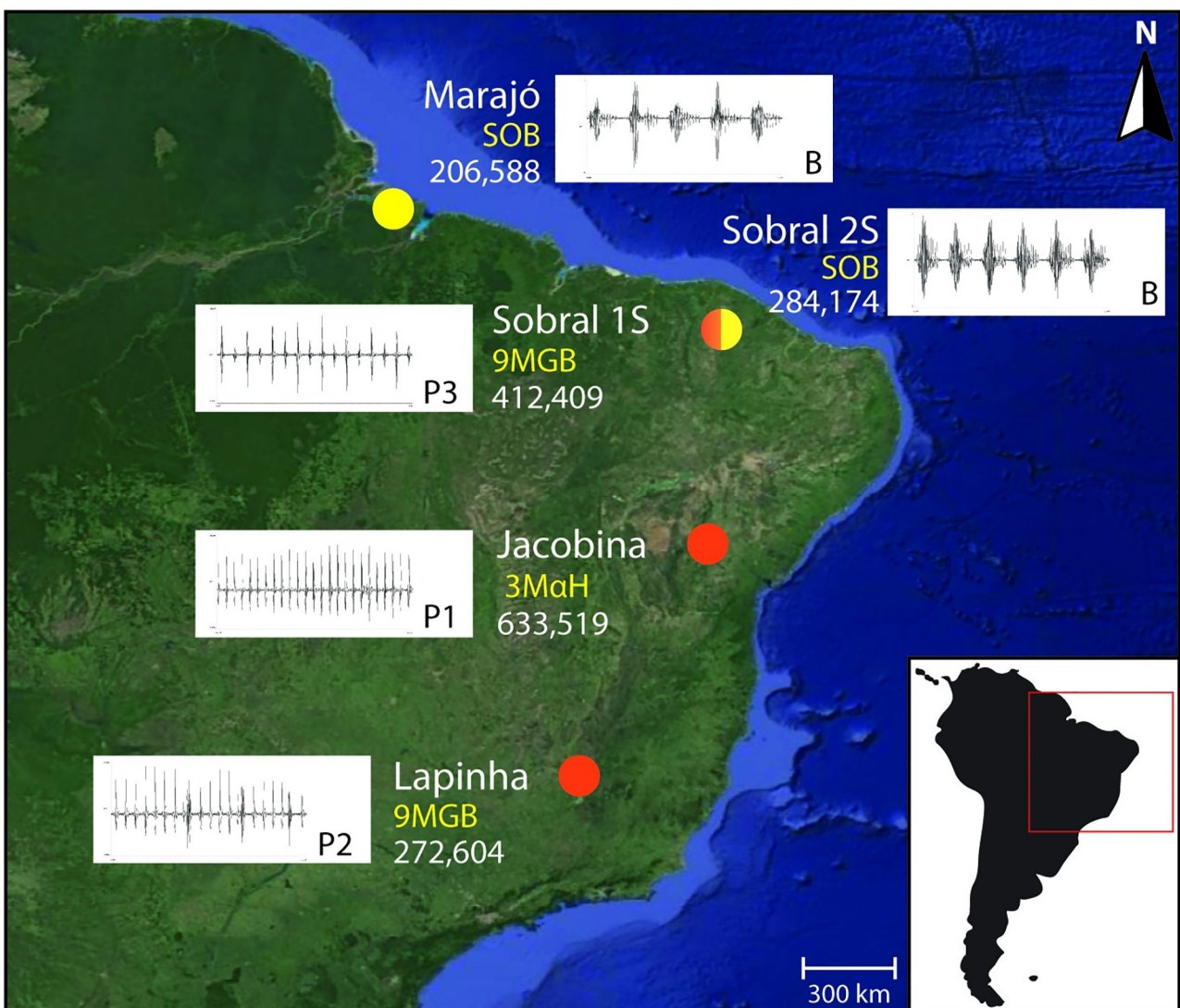

**Fig 1.** ***Lutzomyia longipalpis* site locations for copulation songs and pheromone types.** Samples were collected from three allopatric populations: Marajó (Pará State; 0˚56'S 49˚38'W), Jacobina (Bahia State; $11^0$ 10'S $40^0$ 31'W), and Lapinha Cave (Minas Gerais State; $19^0$ 33'S $43^0$ 57'W); and two sympatric populations from Sobral (Ceará State; $3^0$ 41'S $40^0$ 21'W). Copulation songs: Burst-type (B) and Pulse-types (P1, P2, and P3). Pheromone types: sobralene (SOB), (*S*)-9-methylgermacrene-B (9MGB), and 3-methyl-*α*-himachalene (3M*α*H). For each location, the number of SNPs identified in each population with respect to the reference genome (VectorBase, LlonJ1) is indicated. A total of 1,937,819 SNPs were identified among all the populations. Main map source: World Imagery (Source: Esri, Maxar, Earthstar Geographics, and the GIS User Community; http://goto.arcgisonline.com/maps/World_Imagery). Inset map source: World Dark Gray Canvas Base (Esri, HERE, Garmin, OpenStreetMap contributors, and the GIS user community; http://goto.arcgisonline.com/maps/Canvas/World_Dark_Gray_Base).

nearly 6.8 Mb of sequence was added using PyGap as previously described [55,57–59]. The PyGap program utilizes the Pyramid assembler to detect and merge overlaps of adjoining contigs and closes gaps between non-overlapping adjoining contigs with Illumina data. The same Illumina data used in gap closure was aligned to the assembly to correct 89,378 presumed 454 insertion/deletion errors.

**Lutzomyia longipalpis.** Three types of *Lu. longipalpis* whole-genome shotgun (WGS) libraries were used: a 454 Titanium fragment library and paired end libraries generated from 3 kb and 8 kb inserts. The 454 data (11.5 million reads; ~24.4X coverage) was derived from the

same individual while mate pair reads (7.4 million 3kb reads, 9.6X; 3.7 million 8kb reads, 4.9X) were derived from a pool of individuals. In total, approximately 22.6 million reads were generated at the Baylor College of Medicine Human Genome Sequencing Center (BCM-HGSC) using the Celera CABOG assembler (version 6.1, 2010/03/22) and represents 38.9X coverage of this sand fly genome. These initial results were used as a backbone for longer superscaffolds using ATLAS-link [60]. Finally, discernible gaps were filled (see [61]) with ATLAS-gapfill. The total length of all contigs is 142.7 Mb; however, the total span of the assembly is 154.2 Mb after gaps are included.

## Individual population sequencing

To prepare short insert libraries, an Illumina gel-cut paired-end library protocol was used. Briefly, DNA was extracted from individual adult males or females from inbred lines using the Qiagen DNAeasy Blood and Tissue kit following the manufacturer's supplementary protocol for purification of total DNA from insects. Purified DNA was sheared using a Covaris S-2 system (Covaris, Inc. Woburn, MA). Sheared DNA fragments were purified with Agencourt AMPure XP beads, end-repaired, dA-tailed, and ligated to Illumina universal adapters. After adapter ligation, DNA fragments were further size-selected by agarose gel elution and PCR amplified for 6 to 8 cycles using Illumina P1 and Index primer pair and Phusion High-Fidelity PCR Master Mix (New England Biolabs). The final library was purified using Agencourt AMPure XP beads and quality assessed by Agilent Bioanalyzer 2100 (DNA 7500 kit) determining library quantity and fragment size distribution before sequencing. Sequencing was performed on an Illumina HiSeq2000 platform generating 100 bp paired end reads. Sequenced reads sequence reads were deposited in the NCBI SRA under Bioproject accession number PRJNA20279 for *Lu. longipalpis* and PRJNA20293 for *Ph. papatasi*.

## RNA-sequencing

RNA-sequencing (RNAseq) was conducted to improve resources available for gene prediction. RNAseq was performed following standard protocols on an Illumina HiSeq 2000 platform. To generate an extensive RNA-seq coverage to allow for quality gene prediction, RNA was obtained from both sexes one-, three-, and ten-days post emergence, during development, and females post feeding (6, 24, and 96 hours for *Ph. papatasi* and 6, 24, and 144 hours for *Lu. longipalpis*) on uninfected and *Leishmania* [*Le. major* (MHOM/IL/81/Friedlin) for *Ph. papatasi* and *Le. infantum* (MHOM/BR/76/M4192) for *Lu. longipalpis*] infected mouse blood. Briefly, poly-A$^+$ messenger RNA (mRNA) was extracted from 1 μg total RNA using Oligo(dT)25 Dynabeads (Life Technologies, cat. no. 61002) followed by fragmentation of the mRNA by heating to 94˚C for 3 min [for samples with RNA Integrity Number (RNI) = 3–6] or 4 min (for samples with RIN of >6.0). First-strand complementary DNA (cDNA) was synthesized using the Superscript III reverse transcriptase (Life Technologies, cat. no. 18080–044) and purified using Agencourt RNAClean XP beads (Beckman Coulter, cat. no. A63987). During second-strand cDNA synthesis, deoxynucleoside triphosphate (dNTP) mix containing deoxyuridine triphosphate was used to introduce strand specificity. For Illumina paired-end library construction, the resultant cDNA was processed through end repair and A-tailing, ligated with Illumina PE adapters, and digested with 10 units of uracil–DNA glycosylase (New England Biolabs, Ipswich, MA; cat. no. M0280L). Amplification of the libraries was performed for 13 PCR cycles using the Phusion High-Fidelity PCR Master Mix (New England Biolabs, cat. no. M0531L); 6-bp molecular barcodes were also incorporated during this PCR amplification. These libraries were then purified with Agencourt AMPure XP beads after each enzymatic reaction, and after quantification using the Agilent Bioanalyzer 2100 DNA Chip 7500 (cat. no.

5067–1506), libraries were pooled in equimolar amounts for sequencing. Sequencing was performed on Illumina HiSeq2000s, generating 100-bp paired-end reads. Sequenced reads were deposited in the NCBI SRA, under BioProject accession PRJNA81043 for *Lu. longipalpis* and PRJNA20293 of *Ph. papatasi.*

## Annotation

The genome assemblies were initially annotated *ab initio* with gene models derived from VectorBase annotation MAKER2 [62] pipelines [63]. The automated analyses identified 12,678 gene models for *Ph. papatasi* and 10,429 for *Lu. longipalpis*. Expert curators manually annotated several gene families of interest (S1 Methods) resulting in 11,849 and 10,796 gene models for *Ph. papatasi* and *Lu. longipalpis*, respectively.

## Orthology delineation

OrthoDB [64] orthology delineation was employed to define orthologous groups of genes descended from each last common ancestor of the species phylogeny across 43 insects including the two sand flies—Hemipterodea: *Pediculus humanus* and *Rhodnius prolixus*; Hymenoptera: *Apis mellifera* and *Linepithema humile*; Coleoptera: *Tribolium castaneum*; Lepidoptera: *Bombyx mori* and *Danaus plexippus*; Diptera: *Lu. longipalpis*, *Ph. papatasi* and *Glossina morsitans*, 12 *Drosophila* species (*D. grimshawi*, *D. mojavensis*, *D. virilis*, *D. willistoni*, *D. persimilis*, *D. pseudoobscura*, *D. ananassae*, *D. erecta*, *D. yakuba*, *D. melanogaster*, *D. sechellia*, *and D. simulans*), two culicine mosquitoes (*Aedes aegypti* and *Culex quinquefasciatus*), and 19 *Anopheles* species (*An. darlingi*, *An. albimanus*, *An. sinensis*, *An. atroparvus*, *An. farauti*, *An. dirus*, *An. funestus*, *An. minimus*, *An. culicifacies*, *An. maculatus*, *An. stephensi* (SDA-500), *An. stephensi* (INDIAN), *An. epiroticus*, *An. christyi*, *An. melas*, *An. quadriannulatus*, *An. arabiensis*, *An. merus*, and *An. gambiae* (PEST). The orthology delineation was performed as part of the *Anopheles* Genomes Cluster Consortium analyses of 16 newly-sequenced *Anopheles* mosquitoes [65,66]. From the complete set of species, the two sand flies were compared to a symmetrical set of five representative mosquitoes and five representative flies, together with four outgroup species representing four insect orders. The species compositions of all orthologous groups defined at the dipteran root were analyzed with custom Perl scripts to count the numbers of groups and genes shared among the sand flies, mosquitoes, and flies. Pairwise percent amino acid identities between single-copy and/or multi-copy orthologs among the sand flies, *An. gambiae* and *D. melanogaster* were extracted from all-against-all protein sequence comparisons performed with SWIPE [67] as part of the OrthoDB orthology delineation procedure.

## Maximum likelihood species phylogeny

To establish species relationships, the maximum likelihood species phylogeny was determined from concatenated protein sequence alignments [aligned with default MUSCLE [68] parameters and trimmed with the 'automated1' trimAl [69] setting of 1,627 relaxed single-copy orthologs (no more than three paralogs in up to three species, longest protein selected)] from the two sand flies, five mosquitoes, five flies, and four outgroup insect species. These orthologs were selected from a total of 2,160 orthologous groups and were each required to have an alignment of more than 50 amino acid columns after trimming and a relative tree certainty (see [70]) of more than 50% as implemented in RAxML [71]. The concatenated alignment contained 1,065,440 amino acid columns with 627,808 distinct alignment patterns and was used to estimate the maximum likelihood species phylogeny with RAxML [72] employing the PROTGAMMAJTT model over 100 bootstrap samples and setting *Pe. humanus* as the

outgroup species. The RAxML phylogenies of individual ortholog groups were analyzed with custom Perl scripts and the Newick Utilities [73] to partition the phylogenies into the three relevant topologies—i) sand flies with mosquitoes, ii) sand flies with flies, or iii) sand flies as outgroup to mosquitoes and flies—and all branch lengths were subsequently averaged.

## Population genetics analysis

**SNP calling.** We performed alignments and variant calling on the raw reads of whole-genome samples of *Ph. papatasi* collected from Tunisia (N = 6), Egypt (N = 6), and Afghanistan (N = 5) to the *Ph. papatasi* reference genome (Ppap_1.0). We also aligned and called variants for *Phlebotomus bergeroti* (N = 2), and *Phlebotomus duboscqui* (N = 1) as outgroups. In addition, we called variants from the raw reads of whole-genome samples of *Lu. longipalpis* collected from locations in Brazil [Marajó (N = 9), Lapinha (N = 13), Jacobina (N = 14), and Sobral (9MGB N = 13; SOB N = 16)] and *Nyssomyia intermedia* (N = 2) and *Migonemyia migonei* (N = 2) as outgroups aligned to the *Lu. longipalpis* reference genome (Llon_1.0).

All genomic reads were pre-processed by removing duplicate reads with Picard (v1.113), and paired-end reads were aligned to the reference genome using bwa-mem [74]. Base position differences (SNV) were based on the unique convergence from two variant calling software tools, SAMtools [75] and VarScan 2 [76], using standard variant calling and filtering parameters that are optimized for whole genome data with moderate coverage (10X-40X). These parameters included a *P*-value of 0.1, a map quality of 10, a minimum coverage of three reads, and parameters for filtering by false positives. After alignment and variant detection, we implemented a filter to exclude variants that were clustered in groups of more than five variants per 500 bp. We finally implemented backfilling to include homozygous reference calls for each site where a variant is called in the final multi-sample variant call format (VCF) file for each individual when the coverage exceeded three reads. Sites that did not exceed this threshold were included as missing diploid genotypes.

**SNP filtering.** To aid in the quality assessment of variants, we excluded the genotypes having a genotype quality (GQ) lower than 30 (i.e., minimum accuracy of 99.9%). We also applied hard filters on the variants, excluding any variants having an average depth lower than 10 or higher than 200, a Hardy-Weinberg equilibrium (HWE) *P*-value lower than 0.001, levels of missing genotypes higher than 20%, and having a minor allele frequency (MAF) lower than 1%. The dataset used in population structure inferences was pruned for linkage disequilibrium, excluding variants above an $r^2$ threshold of 0.5 in sliding windows of 50 variants with a step size of 5 variants using PLINK v.1.90 [77,78]. Variants in linkage disequilibrium were pruned from the 6,390,876 sites using a sliding window of 500 kb and a linkage disequilibrium threshold of 0.2 using SNPRelate v.x [79].

**Genomics structure.** Although low powered due to limited sampling, we made an initial attempt to identify regions in the genome that may be contributing to differentiation between the populations. For the *Ph. papatasi* samples, VCFtools v.0.1.15 [80] was used to run a sliding window analysis with a 5 kb sliding window size, a 500 bp step size, and at least 10 variants per window [80]. After calculating Tajima's D for each window within each population [Tunisia (TUN), Egypt (EGP); Afghanistan (AFG)], we calculated pairwise population divergence using Wright's fixation index ($F_{ST}$). We made three pairwise comparisons: i) TUN to EGP; ii) TUN to AFG; and iii) EGP to AFG. The distributions of these results were not normal, so we relied on a percentile approach and selected all 5 kb windows that met the 1st percentile for Tajima's D and the 99th percentile for $F_{ST}$. Windows with fewer than 10 SNPs and windows with coordinates from 1–500 were eliminated. We then searched for 5 kb windows that passed the following thresholds: i) low within-population Tajima's D and ii) high $F_{ST}$. We looked for direct

overlapping windows of high $F_{ST}$ with low Tajima's D scores and indirect overlap, allowing for a 10kb buffer on either end of each window we identified.

Individual ancestry was estimated using Admixture v.1.9 [81]. The analysis was performed for *K* values (ranging from two to seven with 30 iterations per *K*). In order to better understand the different solutions reported by Admixture, post processing of the Admixture results was performed in CLUMPAK v.1.1 [82]. Principal component analysis (PCA) was performed in scikit-allel v.1.1.10 [83], following the methods described in [84]. Weir and Cockerham's $F_{ST}$, Nei's $D_{xy}$, and Tajima's D were calculated using VCFtools, and using the python script pop-GenWindows.py (https://github.com/simonhmartin/genomics_general). Single marker FLK test [85] was performed using HapFLK v.1.4 [86].

**Phylogenetic analysis.** We explored ancestral phylogenetic relationships between individuals by building a neighbor-joining (NJ) tree across the genome using the R packages *adegenet v.2.1.1* [87,88], *ape v.5.1* [89], *poppr v.2.7.1* [90], and *vcfR v.1.7.0* [91]. For *Ph. papatasi*, we included both *Ph. bergeroti* and *Ph. duboscqi* and used the later to root the trees. For *Lu. longipalpis* phylogenetic analysis we included both *N. intermedia* and *M. migonei* and used *M. migonei* to root the NJ trees. We evaluated node support using 1,000 bootstrap replicates [92].

*dN/dS*

Selective constraints on gene sequence evolution were estimated using the dN/dS statistic calculated for orthologous group multiple sequence alignments. Protein sequences were assigned to ortholog groups by cross-referencing the OrthoDB v8 catalog [93]. For ortholog groups with one-to-many and many-to-many orthologs, a single protein sequence was chosen for each species by choosing randomly, with uniform probabilities, from the sequences for each species. Protein sequence multiple alignments were generated first using Clustal-Ω [94], and then used to inform CDS alignments with the codon-aware PAL2NAL alignment program [95]. The yn00 program from PAML v4.8 [96] was used to calculate dN/dS ratios for each pair of sequences in each aligned orthologous group.

# Results and discussion

## Sequencing and genome characteristics

The genome of *Ph. papatasi* is ~350 Mb and was completed in 2012 for community analysis and population comparisons (S1 Table). The assembly was built from an input of ~22.5X total sequence coverage and resulted in 139,199 contigs with an N50 of 5.8 kb and 106,826 scaffolds with an N50 of 28 kb. The draft genome of *Lu. longipalpis* (Llon_1.0) was also completed in 2012 and is approximately 154.2 Mb, more than two times smaller than the *Ph. papatasi* genome, representing 38.9X coverage (S1 Table). There are 35,696 contigs with an N50 of 7.5 kb and 11,532 scaffolds with an N50 of 85.1 kb. Based on automated and manual annotations, the *Ph. papatasi* and *Lu. longipalpis* genomes are estimated to contain 11,216 and 10,311 protein-coding genes, respectively. The BUSCO analysis [97] indicated 86.5% and 86.1% completeness for the *Ph. papatasi* and *Lu. longipalpis* genomes respectively (S2 Table). The N50 sizes and BUSCO completeness scores suggest that the assemblies are fragmented and may be missing regions of the genomes. Annotation was augmented with RNA-seq expression evidence from different life-cycle stages, multiple days post adult emergence, and after blood-feeding in uninfected and *Le. major*-infected blood for *Ph. papatasi* and *Le. infantum*-infected blood for *Lu. longipalpis* (S3 Table).

Orthology

To improve our understanding of the phylogenetic relationships, we generated a maximum likelihood phylogenetic tree using orthologous genes selected from an orthology dataset comprising 43 insect species, including 36 dipterans. Consistent with [14], the phylogenetic tree

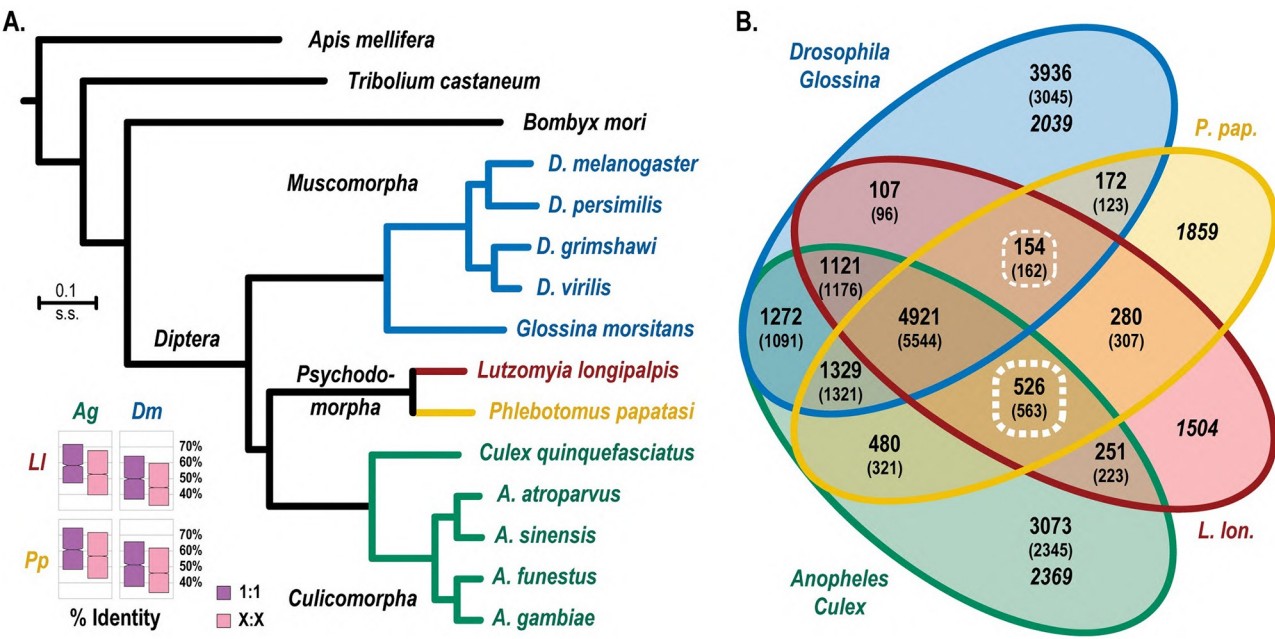

**Fig 2. Molecular species phylogeny and ortholog sharing.** (A) The quantitative maximum likelihood species phylogeny computed from the concatenated superalignment of 1,627 orthologous protein-coding genes places the sand flies (Psychodomorpha) as a sister group to the mosquitoes (Culicomorpha) rather than the flies (Muscomorpha), with all branches showing 100% bootstrap support. The Culicomorpha are represented by four *Anopheles* mosquito species and *Culex quinquefasciatus* and the Muscomorpha include four *Drosophila* fruit fly species and the tsetse fly, *G. morsitans*. Outgroup species represent Lepidoptera (*Bombyx mori*), Coleoptera (*T. castaneum*), Hymenoptera (*Apis mellifera*), and the phylogeny is rooted with the phthirapteran human body louse, *Pe. humanus*. The inset boxplots show that single-copy (1:1) and multi-copy (X:X) ortholog amino acid percent identity is higher between each sand fly (Ll, *Lu. longipalpis*; Pp, *Ph. papatasi*) and *An. gambiae* (Ag) than *D. melanogaster* (Dm). Boxplots show median values with boxes extending to the first and third quartiles of the distributions. (B) The Venn diagram summarizes the numbers of orthologous groups and mean number of genes per species (in parentheses) shared among the two sand flies (L. lon., *Lu. longipalpis*; P. pap., *Ph. papatasi*) and/or the Culicomorpha and/or the Muscomorpha. Analysis of ortholog sharing shows that the sand flies share more than three times as many orthologous groups exclusively with the Culicomorpha (*Anopheles* and *Culex*) compared to the Muscomorpha (*Drosophila*, *Glossina*) (subsets highlighted with thin and thick dashed lines). Numbers of unique genes are in italics. Colors in panel A and panel B match species and sets of species analyzed.

supported clustering of sand flies with Culicomorpha infraorder (mosquitoes and black flies) rather than with the Muscomorpha infraorder (*Drosophila* and *Glossina)* (Fig 2A). In addition, percent identity between orthologs is higher between sand flies and mosquitoes than between sand flies and fruit files, in agreement with the maximum likelihood phylogeny (Fig 2A). Sand flies and culicines have more than three times as many exclusively-shared orthologous groups than sand flies do with muscoids, also consistent with the maximum likelihood phylogeny (Fig 2B). Analysis of individual gene phylogenies, however, shows great uncertainty with almost equal proportions of phylogenies supporting clustering of sand flies with mosquitoes and with muscoids (S1 Fig).

**Transposable elements.** Transposable elements (TEs) are ubiquitous repetitive sequences present in eukaryotic genomes that can be an important factor affecting genome sizes and are thought to be one of the driving forces of evolution [63]. Some insect genomes have less than 3% of TEs, while others contain as much as 50% or more of TEs, associated with large genomic size differences [98]. Our analysis indicated that the *Ph. papatasi* genome is composed of 5.65% of TE derived sequences while the *Lu. longipalpis* genome contains only 0.57%, corresponding to the genome size difference between two sand fly species. This difference in TE-derived sequence could be due to the result of divergent evolutionary dynamics of some TE families or superfamilies, affecting either their distribution (presence or absence of specific

superfamilies) or their abundance (copy number per superfamily) in the genome. Higher abundance of TE derived sequences, presence of full-length TEs and the genome size expansion in the *Ph. papatasi* genome also could be due to recently active TEs. Alternatively, genomic differences in TE content might be the result of intrinsic genomic deletion patterns in *Lu. longipalpis*, due to the effective recognition and elimination machinery removing these foreign sequences from the genome, as has been shown to occur in *Drosophila* species [99].

Although the fragmented nature of the genome assemblies makes a completely accurate assessment of TE content difficult, the comparison of the TE content in both sand fly genome assemblies suggest an expansion of all the TE classes and orders in the *Ph. papatasi* genome. This multiplication was more pronounced in elements belonging to the class II, or "cut-and-paste" TEs, and especially in non-autonomous miniature inverted-repeat transposable elements (MITEs), representing up to 29-fold differences between the two genomes. Expansion of MITEs suggests the recent activity of class II TEs in the *Ph. papatasi* genome. On the other hand, class I elements, or "copy-paste" elements, including the Long Terminal Repeat (LTRs) and non-LTRs, which traditionally are accountable for the genome expansion, showed more subtle changes between the two sand fly genomes, representing up to 4-fold difference. (Table 1).

**Immunity genes.** Several immune pathways are conserved among insects. These include the Toll, Immune Deficiency (IMD), Janus Kinase/Signal Transducer and Activator of Transcription (JAK/STAT), lectin, and encapsulation pathways. We found the Toll signaling pathway highly conserved in the genomes of both sand fly species, including homologues of the upstream peptidoglycan recognition proteins (PGRPs), and glucan binding protein (GNBPs) (S4 Table). Similarly, the IMD and JAK/STAT pathways appear to be conserved among dipterans, including *Drosophila*, *Aedes*, *Anopheles* and both sand fly species analysed in this study (S5 and S6 Tables).

Galactose-binding proteins (galectins) are a diverse family of proteins playing roles in development and immunity [100]. Comparing the sand flies' galectin protein sequences with other Diptera, both shared and independent orthologs were identified (S2 Fig and S7 Table). Future analyses evaluating *Leishmania* parasite interactions with the *Ph. papatasi* and *Lu. longipalpis* galectins may provide a better understanding of the mechanisms that influence restrictive versus permissive vectorial competence due to the key role some galectins play in parasite establishment and survival [101].

Fourteen genes related to TGF-beta or TGF-beta pathways were found in each of the sand fly genomes (S8 Table) and 16 and 15 different MAPK gene loci were identified in the genome of *Lu. longipalpis* and *Ph. papatasi* genome, respectively (S9 Table). Interestingly, two prophenoloxidase homologues were identified in each species (S10 Table). A TEP-1-like protein, and a COX-like ortholog were also found in the genomes of both *Lu. longipalpis* and *Ph. papatasi* (S5 Table).

**Blood feeding genes.** We mapped *Ph. papatasi* and *Lu. longipalpis* putative salivary genes deposited at the NCBI to the sand fly assemblies (S11 Table). Equally well studied in phlebotomine sand flies are the genes associated with digestive properties [102–111]. Here, we characterized the following digestive gene families: Peptidases (S12 Table); Glycoside Hydrolase Family 13 (S13 Table); Chitinase and Chitinase-like protein family (S14 Table); N-acetylhexosaminidases (S15 Table and S3 Fig); Chitin deacetylases (S16 Table and S4 Fig); and Peritrophin-like proteins (S17 Table and S5 Fig). A detailed analysis of GH13 genes, including amylases, maltases and sucrases has been published elsewhere [112]. Aquaporins (AQPs) are required for the transportation of water and other small solutes across cell membranes and are important for excreting water from the blood meal. We have identified six AQP genes from both species of sand flies (S18 Table and S6 Fig). This is similar to the number present in

**Table 1. Transposable Elements.**

|  | *Phlebotomus papatasi* | *Lutzomyia longipalpis* |
|---|---|---|
|  | % genome | % genome |
| **LTR retrotransposons** | **0.41%** | **0.21%** |
| Bel | 0.17% | 0.09% |
| Mag | 0.07% | 0.04% |
| Pao | 0.06% | 0.01% |
| Mdg3 | 0.05% | 0.02% |
| Gypsy | 0.03% | 0.02% |
| Mdg1 | 0.02% | 0.01% |
| Osvaldo | 0.01% | 0.01% |
| Copia | 0.01% | 0.00% |
| **Non-LTR retrotransposons** | **0.95%** | **0.22%** |
| L2 | 0.25% | 0.04% |
| RTE | 0.21% | 0.02% |
| Jck | 0.18% | 0.04% |
| CR1 | 0.16% | 0.03% |
| LOA | 0.07% | 0.02% |
| I | 0.05% | 0.05% |
| Loner | 0.03% | 0.01% |
| Ocas | 0.01% | 0.01% |
| **DNA transposons** | **1.13%** | **0.05%** |
| Tc1/mariner | 0.96% | 0.04% |
| piggyBac | 0.18% | 0.00% |
| Helitron | 0.00% | 0.01% |
| **MITEs** | **4.11%** | **0.14%** |
| mTA | 2.63% | 0.01% |
| m2bp | 0.54% | 0.01% |
| m8bp | 0.29% | 0.10% |
| m3bp | 0.24% | 0.00% |
| otherMITEs | 0.23% | 0.00% |
| m4bp | 0.18% | 0.03% |
| **Unclassified TE sequences** | **0.00%** | **0.01%** |
| **Total, percent TE in genome** | **5.65%** | **0.57%** |

mosquitoes (N = 6), but two and four less than *Drosophila* and *Glossina*, respectively [113]. Members of each AQP group previously identified from insects are present in the sand fly genomes.

**Circadian rhythm genes.** Orthologues of all the core circadian clock genes were found in the genome of both *Lu. longipalpis* (S19 Table) and *Ph. papatasi* (S20 Table). Interestingly, cryptochrome evolution has been a matter of great interest [48] and we similarly found compelling features in sand fly CRY gene structure. We found both *CRY1 and CRY2* genes in *Ph. papatasi* but, surprisingly, we did not find a *CRY1* gene in *Lu. longipalpis* genome assembly (S7 Fig). Although both sand fly species are closely related, these data suggest that whereas *Ph. papatasi* seems to have a functional mammalian-like clock closer to butterflies, mosquitoes, and other dipterans, with *CRY1* and *CRY2* genes, *Lu. longipalpis* may have a circadian clock working with a mechanism more similar to that found in triatomines, bees and beetles, presenting only *CRY2* gene. We can speculate that the possible loss of *CRY1* in *Lu. longipalpis*

genome may be related to a better adaptation of these insects to living in caves and dark places or alternatively, is just missing in the current fragmented assembly.

**Chemosensory receptors.** The sand fly olfactory receptor (OR), gustatory receptors (GR), and ionotropic receptor (IR) repertoires were published elsewhere [35]. The sand fly OR repertoires in the genome assemblies comprise 139 canonical ORs in *Lu. longipalpis* and *Ph. papatasi*, plus one copy each of the odorant receptor co-receptor, *Orco*. Eighty-two genes encoding 91 GRs in *Lu. longipalpis* and 77 genes encoding 88 GRs in *Ph. papatasi* were identified in the reference assemblies, and 23 and 28 IR genes in *Lu. longipalpis* and *Ph. papatasi*, respectively were identified. Three ORs and three IRs suspected to be missing in the *Lu. longipalpis* reference assembly were found in *de novo* assemblies of the field isolates [35].

Nine and ten members of the transient receptor potential (TRP) cation channel family are found in *Lu. longipalpis* and *Ph. papatasi* genomes, respectively, and the phylogenetic tree showed a separation of the different TRP subfamilies (S8 Fig). The pickpocket (PPK) receptor phylogenetic tree demonstrated a division of the six different PPK subfamilies (S9 Fig) with 14 and 13 family members in *Lu. longipalpis* and *Ph. papatasi* genomes, respectively.

**G-Protein coupled receptors.** G-Protein Coupled Receptors (GPCRs) are a large family of membrane-bound proteins that operate in cellular signal transduction and interact with a wide variety of chemistries including small molecules, neuropeptides, and proteins. These proteins play roles in essential invertebrate functions [114]. We utilized a novel classifier to identify insect GPCRs [115] in both *Ph. papatasi* and *Lu. longipalpis*, followed by validation and manual annotation of identified genes. Ninety-four and 92 GPCRs from *Ph. papatasi* and *Lu. longipalpis*, respectively, were compared with other insects with well characterized GPCRs such as *D. melanogaster*, *An. gambiae*, *Ae. aegypti* and *Pe. humanus* (S21 Table). Class A (rhodopsin-like) is the most numerous class with ~50 genes in each sand fly, and includes the opsins that are thought to function in visual processes and circadian rhythm. Both sand flies have one opsin gene for each functional group, the long-wavelength, short-wavelength, ultraviolet, rh7-like, and pteropsin. Classes B (secretin-like), C (metabotropic glutamate-like) and D (atypical GPCRs) have fewer members, with ~20, ~10 and ~10 in each sand fly, respectively. Sand flies include GPCR genes absent from *D. melanogaster* (ocular albinism) and absent in *Ae. aegypti* and *An. gambiae* (parathyroid hormone receptor); both genes from class B.

**Cytochrome P450 monoxygenase genes.** Cytochrome P450s (CYPs or P450s) constitute a conserved enzyme superfamily with a diverse array of functions, ranging from core developmental pathways to the detoxification of xenobiotics [116]. The CYP gene repertoire (CYPome) plays an important role in insect physiology and in the development of resistance to insecticides used for vector control. Here we identified and manually curated 104 CYP genes in *Lu. longipalpis* (S1 Data) and 93 CYP genes in *Ph. papatasi* (S2 Data). These numbers are similar to the number of CYPs identified in the mosquito *An. gambiae* (n = 100). In *Lu. longipalpis* all 104 CYPs are full-length genes, compared to 34 full-length and 59 fragmented genes in *Ph. papatasi*, likely reflecting the more fragmented genome assembly of *Ph. papatasi* compared to *Lu. longipalpis*.

The identified sand fly CYP genes belong to the four clans typically found in insects; mitochondrial (Mito), CYP2, CYP3, and CYP4 clan [116]. Remarkably, both sand fly species have an expanded CYP3 clan compared to *An. gambiae* (S10 Fig). This expansion is mostly caused by gains in the CYP9J/9L, CYP6AG, and CYP6AK subfamilies (S10 Fig).

**Other groups.** In addition, we identified core genes as well as non-coding RNAs in the siRNA, miRNA, and piRNA pathways, suggesting that these regulatory mechanisms are fully functional in sand flies (S22 Table). We have also annotated heat shock and hypoxia proteins (S23 Table), cuticular proteins (S24 Table), hormonal signaling (S25 Table), insulin signaling

(S26 Table), and antioxidant (S27 Table) genes, as well as genes involved in vitamin metabolism (S28 Table). Additional information about annotated gene families can be found in the S1 Results.

## Population structure

**Genetic structure across the range of *Ph. papatasi*.**   Average genome coverage ranged from 8X-16X (mean = 12X; S29 Table). A total of 6,390,876 sites passed the thresholds using variant calling methods, where at least one sample displayed a variant at a reference coordinate. As expected, the *Ph. papatasi* samples showed the lowest count of Single Nucleotide Variants (SNVs) (1.84–1.99M SNVs) while the two *Ph. bergeroti* samples (mean SNVs = 3.26M), and the *Ph. duboscqi* sample (4.01M SNVs) contained a higher variant count (S30 Table). We found a small percentage of singletons (unique SNV's) in the *Ph. papatasi* samples (3.0%-4.3%) and 3,482 shared variant alleles among the *Ph. papatasi* samples. We also calculated the transition: transversion ratios, inbreeding coefficients, and pairwise relatedness (S30 Table).

For phylogenetic analysis the dataset was filtered by keeping only variants of the highest quality, leaving 1,084,952 total variants: 284,341 for Afghanistan, 435,972 for Egypt, and 439,446 for Tunisia. The dataset used in population structure inferences was further pruned for linkage disequilibrium, creating a final dataset containing 423,236 total variants.

We explored ancestral phylogenetic relationships between individuals by building a NJ tree across the genome. The NJ tree clustered the *Ph. papatasi* individuals into three clades that correlated to geographical location, with bootstrap values of 100 (S11A Fig).

Admixture was used to estimate the individual ancestries. Admixture cross-validation errors (CV) suggest that the number of genetic clusters that best explains the observed population structure as $K = 2$ (S12A Fig), where the Afghanistan samples were distinct from the Tunisian and Egyptian samples (S11C Fig).

We next performed a PCA, which does not depend on any model assumption and can thus provide a useful validation of the results of Admixture analysis. The PCA supported the phylogenetic analysis, separating the individuals into three distinct clusters, with all individuals from the collection site clustering together. Principal components 1 and 2 accounted for 20.1% of the total variation (S11B Fig).

We found no direct overlapping windows of high $F_{ST}$ with low Tajima's D scores for any of the *Ph. papatasi* populations. Next, we searched for windows that met the above criteria but included a 20 kb (10 kb on either side of the window) to identify indirect overlaps. We identified 29 genes that fell within in the indirect overlapping windows (S31 Table). Functional annotation revealed 3 tRNAs, 3 putative transcription factors, and a snoRNA as possibly under selective pressure, as well as 9 genes involved in metabolic pathways.

**Genomic evidence of cryptic species within *Lu. longipalpis sensu lato*.**   The average genome coverage ranged from 8X to 105X (mean = 47X; S32 Table). We identified 4,821,847 variants across all the individuals. To aid in quality assessment of variants, filtration was performed as described for *Ph. papatasi*. After filtration, 1,937,819 variants remained, ranging from 206,588 for Marajó to 633,519 for Jacobina (Fig 1). After filtration and LD pruning, the dataset contained 1,059,627 variants.

Consistent with the phylogeny based on the chemoreceptor repertoire [35], the full genome phylogeny clustered the populations into two clades based on song and pheromone type, where Marajó and Sobral 2S (Burst, Sobralene) cluster together and Lapinha and Sobral 1S (Pulse, (*S*)-9-methylgermacrene-B) cluster together (Fig 3A). An analysis of the male copulatory courtship songs of males collected from Lapinha and Sobral are in agreement with those

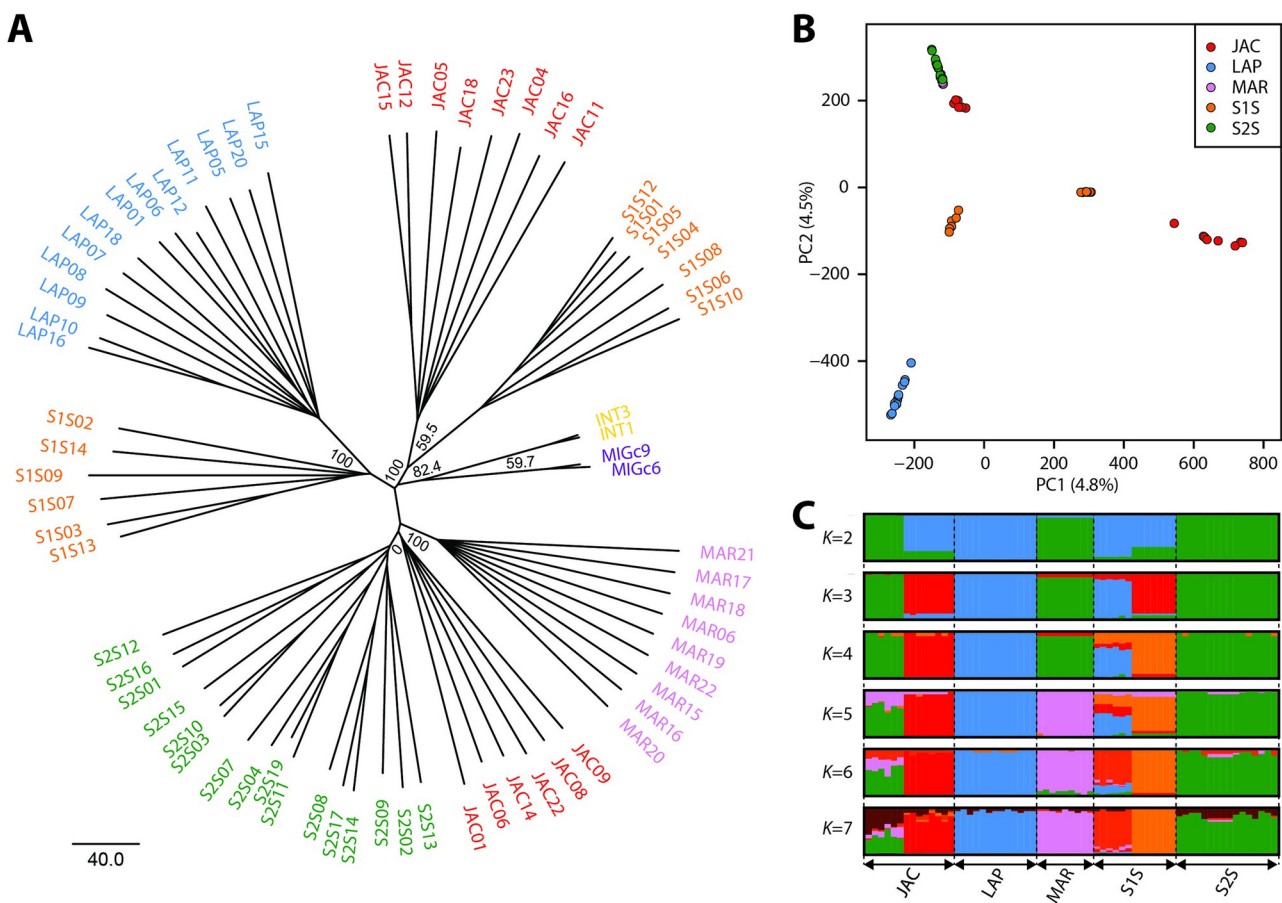

**Fig 3. *Lutzomyia longipalpis* population structure.** Inferred population structure of *Lu. longipalpis* individuals collected from Marajó (MAR; pink), Lapinha (LAP; blue), from Jacobina (JAC; red), and Sobral, including Sobral 1S (S1S; orange) and 16 Sobral 2S (S2S; green). (A) Rooted neighbor joining (NJ) radial tree. We included both *N. intermedia* (INT; yellow) and *M. migonei* (MIG; purple) and used *M. migonei* to root the trees. Bootstrap values represent the percentage of 1,000 replicates. (B) Principal component analysis (PCA). Individuals were plotted according to their coordinates on the first two principal components (PC1 and PC2). (C) Admixture analysis. Ancestry proportions for Admixture models from $K = 2$ to $K = 7$ ancestral populations. Each individual is represented by a thin vertical line, partitioned into $K$ coloured segments representing the individual's estimated membership fractions to the $K$ clusters. These data are the average of the major $q$-matrix clusters derived by CLUMPAK analysis.

previously recorded [27]. In these three resampled populations, we observed the sub-type P2 in Lapinha, the sub-type P3 in Sobral 1S, and the burst-type in Sobral 2S.

Interestingly, the phylogenetic analysis separated the Jacobina population into two groups. As expected, because flies from Jacobina are known to express a $C_{16}H_{32}$ pheromone and pulse-like copulatory songs, some individuals clustered with Sobral 1S and Lapinha. Unexpectedly, however, six individuals clustered with the diterpenoid-like pheromone and burst song expressing individuals, suggesting that there is more than one population living in sympatry at the Jacobina site. Male copulatory songs were not recorded for the Jacobina samples. However, sand flies collected from two localities near to Jacobina, Araci and Olindina (Bahia state) (S13A Fig) exhibit different copulatory song patterns, suggesting the possible existence of two groups in Jacobina, as observed in molecular data. Males from Araci exhibit the P1 copulation song pattern (S13B Fig), composed of train of similar pulses as previously described in males collected in Jacobina [27]. Males from the nearby locality, Olindina, produced burst-type songs (S13B Fig) with similar pattern as Sobral 2S males [27]. The mean values of all song parameters observed from these flies (S33 Table) are similar as previously reported [29].

In addition, the phylogenetic analysis indicated sub-structure within the Sobral 1S population. The song tracings, however, did not suggest any sort of split. Although the analysis suggests seven distinct populations, there is not enough statistical support to separate the six Jacobina individuals from the Sobral 2S population, resulting in support for six populations.

The PCA based on the whole-genome clustered the individuals into six groups as well (Fig 3B). Contrary to the phylogenetic analysis, however, the six Jacobina individuals were closely clustered, but separate from the Marajó and Sobral 2S populations, which were indistinguishable. PC1 explained 5.3% of the variation and separated the individuals collected from Jacobina into two populations. Consistent with the NJ tree, the Sobral 1S population also exhibited some population structure, the two clusters distinguishable through PC1 and PC2. PC2 accounted for 4.6% of the total variation and distinguished Lapinha from the other populations. The sympatric Sobral 1S and 2S populations separate by both PC1 and PC2. Interestingly, while consistent with Hickner *et al.* 2020, the whole-genome PCA allowed higher discriminating power among clusters than the PCA based on the chemoreceptor repertoire which only identified 3–4 discrete clusters [35].

Seven groups are clearly distinguishable from the Admixture analysis at $K = 7$, consistent with the PCA, NJ tree (Fig 3C), and [35]. However, the cross-validation error analysis indicates 3–4 populations (S12B Fig), one population consisting of all Marajó and Sobral 2S individuals and six Jacobina individuals, one population made up of 8 Jacobina individuals and another population with 7 Sobral 1S individuals. In contrast to the NJ tree that suggests that the individuals from Lapinha make up a single population, the Admixture analysis indicates that all Lapinha individuals and six Sobral 1S individuals are of similar ancestry. The analysis suggests no introgression between the sympatric Sobral 1S and 2S individuals.

To identify candidate genomic regions contributing to reproductive isolation and to distinguish between the two models of speciation, that with and without gene flow, pairwise measures of divergence were calculated for Marajó, Sobral 1S, Sobral 2S, and Lapinha. Relative (Weir and Cockerham's $F_{ST}$) and absolute (Nei's $D_{xy}$) measures of divergence were calculated for 1 kb non-overlapping windows for all population comparisons, excluding Jacobina. Mean weighted $F_{ST}$ values indicate that genome wide differentiation is greater among population comparisons of different pheromone and song types (Lapinha- Marajó, 0.214; Lapinha-Sobral 2S, 0.211; Sobral 1S-Sobral 2S, 0.116 compared to Lapinha—Sobral 1S, 0.154; Marajó-Sobral 2S, 0.114) and allopatric populations (Lapinha- Marajó, 0.214; Lapinha-Sobral 2S, 0.211; Lapinha—Sobral 1S, 0.154 compared to Sobral 1S-Sobral 2S, 0.116) (S14 Fig).

We identified genomic regions possibly contributing to population differentiation as $F_{ST}$ outlier windows that were in the top 2.5% quantile for each sympatric and allopatric comparison of differing pheromone/song phenotype (S15 Fig). There were 170 differentiation regions in common among all of the different pheromone and song type comparisons (Fig 4A). The mean $F_{ST}$ estimates were higher in the genomic regions shared by more than one comparison than in those unique to each comparison, suggesting that these regions are being targeted by selection in each case. Supporting the hypothesis that the Sobral populations have more recently diverged from one another, the $F_{ST}$ outlier windows had a mean value less than the allopatric populations (Fig 4B).

We further characterized the genomic regions by computing additional statistics in each window. We tested if these regions were enriched for signatures of selection by computing Tajima's D in the 1 kb non-overlapping windows, negative values of Tajima's D indicating a potential selective sweep. As with the $F_{ST}$ values, we considered outlier windows as those that were in the lower or upper 2.5% quantiles (S16 Fig). The vast majority of Tajima's D outlier windows were unique to each population (S17 Fig). No positive outlier windows overlapped among the four populations (S17A Fig) and only four negative outlier windows were shared

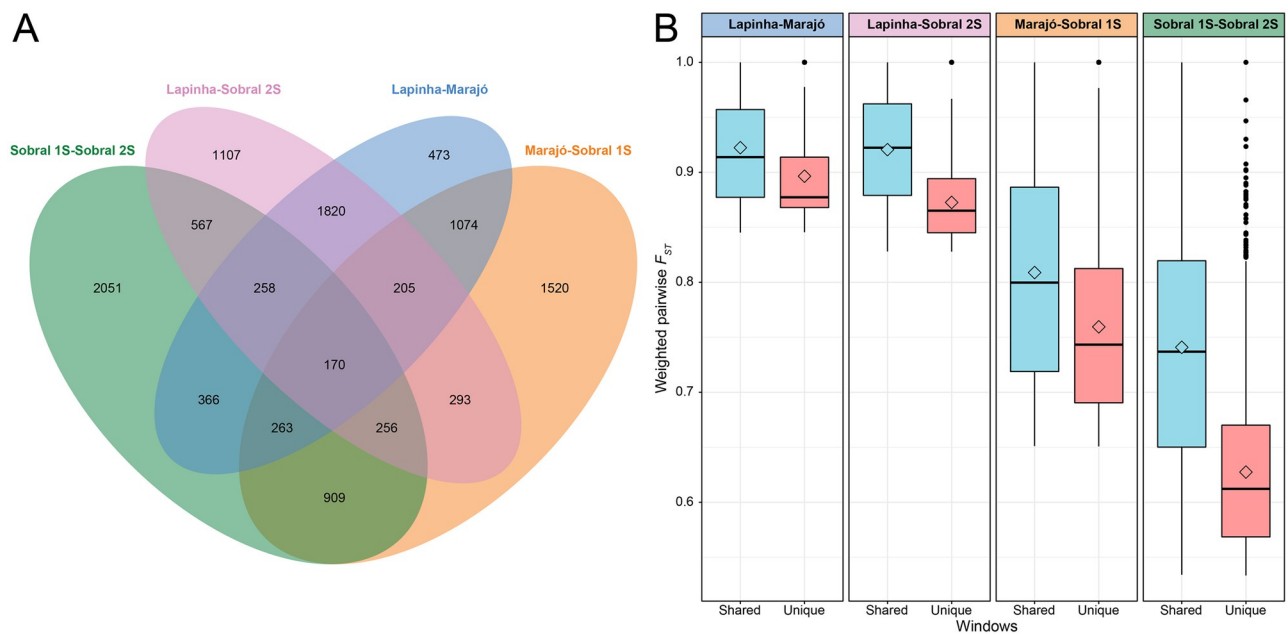

**Fig 4. Genomic regions with high pairwise $F_{ST}$ between the different populations of *Lutzomyia longipalpis*.** (A) Venn diagram depicting the number of 1 kb non-overlapping genomic windows having $F_{ST}$ values in the top 2.5% quantile (outlier) among the different population comparisons. (B) Box plots of outlier $F_{ST}$ windows shared with another population comparison (blue) or unique to a population comparison (pink). Box plots show the medians (lines) and interquartile ranges (boxes); the whiskers extend out from the box plots to 1.5 times the interquartile range, and values outside this limit are represented by dots. Mean $F_{ST}$ values are represented by open diamonds.

among all of the populations (S17B Fig), of which only one contained a gene, LLOJ005792 of unknown function. None of the Tajima's D outlier windows overlapped with the outlier $F_{ST}$ windows.

As absolute measures of divergence are less affected by within population levels of polymorphism than relative measures of divergence, like $F_{ST}$ [117], we calculated Nei's measure of absolute divergence, $D_{xy}$, as an additional signature of selection. As expected, because these populations are thought to have recently diverged from one another, the top 2.5% of $D_{xy}$ values were substantially lower than the outlier $F_{ST}$ windows (Fig 5A). The majority of outlier $F_{ST}$ values did not fall in the upper quantile of $D_{xy}$ values (Table 2) and the windows with the highest $D_{xy}$ values did not overlap with the $F_{ST}$ outlier windows (Fig 5B), suggesting that there may be varying levels of genetic diversity within each population.

To identify genomic loci that may be contributing to the reproductive isolation of these populations [39], we defined 'regions of interest' as those windows that fell in both the upper 2.5% quantile of $F_{ST}$ and $D_{xy}$ values. There were 729, 841, 740, and 1023 regions of interest between Lapinha-Marajó, Lapinha-Sobral 2S, Marajó-Sobral 1S, and Sobral 1S-Sobral 2S, respectively (Table 2). The 92 regions of interest shared among all the population comparisons we interpreted as 'differentiation islands' (DI).

We tested whether the DIs were enriched for signatures of selection by calculating Tajima's D for these windows and performing a single marker FLK test [85] with HapFLK v. 1.4 [86]. The Tajima's D (Fig 5C) and FLK (Fig 5D) values do not provide evidence that selection (either balancing or positive) has led to the genomic divergence in these regions.

The genes present in the DIs are candidates that might explain the reproductive isolation of the populations. The 92 DIs contained 35 genes, 25 of which had orthologues in *An. gambiae* (S34 Table). Thirty-two of these genes were uncharacterized, LLOJ001208 is a protein MAK16

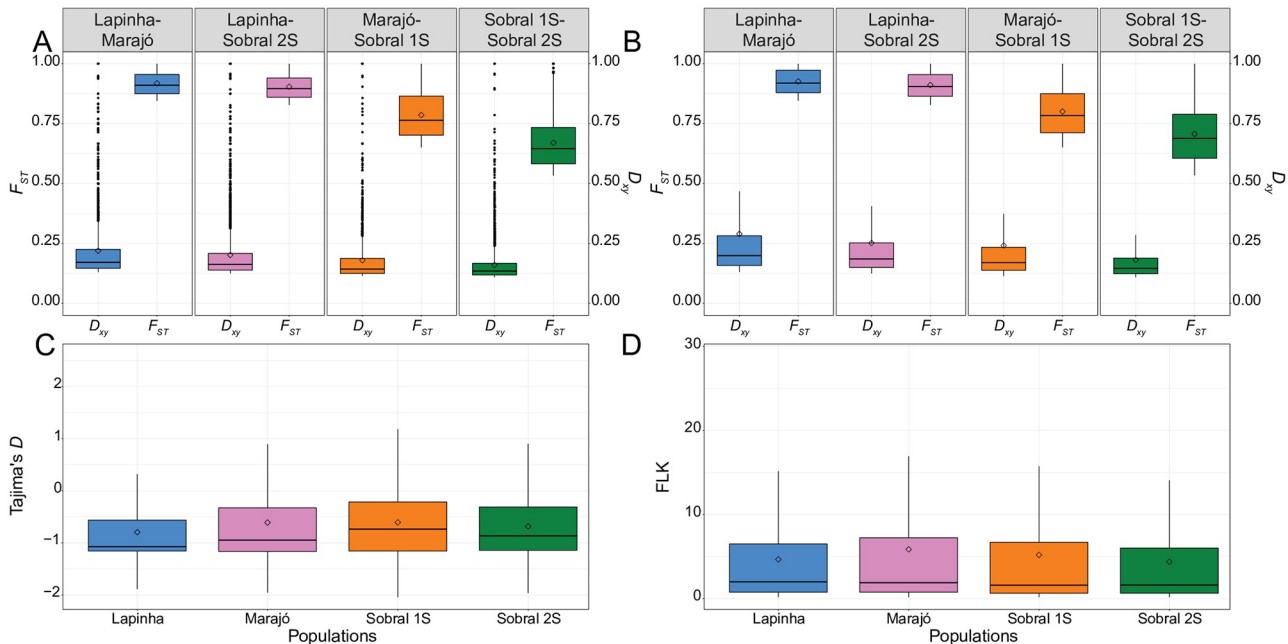

**Fig 5. Measures of divergence in 1 kb non-overlapping genomic windows between the different populations of *Lutzomyia longipalpis*.** (A) Box plots of $D_{xy}$ and $F_{ST}$ values in the top 2.5% quantile (outlier) of each population comparison. (B) Box plots of $D_{xy}$ and $F_{ST}$ values for windows having both high $D_{xy}$ and high $F_{ST}$ (differentiation islands). (C) Box plots of Tajimas' D values for the differentiation islands. (D) Box plots of FLK values for sites within differentiation islands. Box plots show the medians (lines) and interquartile ranges (boxes); the whiskers extend out from the box plots to 1.5 times the interquartile range, and values outside this limit are represented by dots. Mean values are represented by open diamonds.

homolog, LLOJ009447 a rRNA adenine N(6)-methyltransferase, and LLOJ009732 a Lipase maturation factor. No enrichment of gene ontology terms was identified using the *An. gambiae* orthologs.

## Conclusions

Our study provides the genome assembly and annotation of two divergent sand fly species that will facilitate molecular and comparative studies of these medically important vectors. These results provide a foundation for annotating and analyzing future chromosome length assemblies generated from single sand flies. Global comparisons between sand fly vectors will greatly inform the evolutionary relationships among these species and lead to advances in our

**Table 2. *Lutzomyia longipalpis* Differentiation Island (DI) Statistics.**

| | Lapinha- Marajó | Lapinha-Sobral 2S | Marajó-Sobral 1S | Sobral 1S-Sobral 2S |
|---|---|---|---|---|
| **# 1 kb Windows** | 127,065 | 129,513 | 127,065 | 131,430 |
| **# FST Outlier Windows** | 3,176 | 3,237 | 3,176 | 3,285 |
| **# Regions of Interest** | 729 | 841 | 740 | 1,023 |
| **% FST Outlier Non-DI** | 77.05 | 74.03 | 76.70 | 68.87 |
| **Mean $F_{ST}$ DI** | 0.93 | 0.85 | 0.85 | 0.86 |
| **Mean Dxy DI** | 0.30 | 0.29 | 0.29 | 0.29 |

There were 92 1 kb windows that fell in the upper 2.5% of $F_{ST}$ and $D_{xy}$ values and shared among all the comparisons. These windows were defined as Differentiation Islands (DI).

understanding of genes involved in important phenomena such as vectorial capacity, host-specificity, blood-feeding, insecticide resistance, and immune system modulation.

## Supporting information

**S1 Methods. Methods used for manual annotation.**
(DOCX)

**S1 Results. Detailed information of annotated gene families.**
(DOCX)

**S1 Table. Assembly statistics.**
(DOCX)

**S2 Table. BUSCO analysis.**
(DOCX)

**S3 Table. RNAseq samples.**
(XLSX)

**S4 Table. Toll pathway annotation.**
(XLS)

**S5 Table. Insect immune deficiency pathway annotation.**
(XLSX)

**S6 Table. JakStat pathway annotation.**
(XLSX)

**S7 Table. Galectin family annotation.**
(XLS)

**S8 Table. Transforming growth factor-beta family annotation.**
(XLSX)

**S9 Table. Mitogen activated protein kinase family annotation.**
(XLSX)

**S10 Table. Prophenoloxidase family annotation.**
(XLSX)

**S11 Table. Salivary protein annotation.**
(XLS)

**S12 Table. Peptidase annotation.**
(PDF)

**S13 Table. Glycosidase Hydrolase family 13 annotation.**
(XLSX)

**S14 Table. Chitinase family annotation.**
(XLSX)

**S15 Table. Hexosaminidase family annotation.**
(XLS)

**S16 Table. Chitinase deacetylase family annotation.**
(XLS)

**S17 Table. Peritrophin family annotation.**
(XLS)

**S18 Table. Aquoporin family annotation.**
(XLSX)

**S19 Table.** *Lutzomyia longipalpis* **circadian rhythm pathway annotation.**
(DOCX)

**S20 Table.** *Phlebotomus papatasi* **circadian rhythm pathway annotation.**
(DOCX)

**S21 Table. G-protein coupled receptor family annotation.**
(XLSX)

**S22 Table. MicroRNA annotation.**
(XLS)

**S23 Table. Heat shock and hypoxia gene family annotation.**
(XLSX)

**S24 Table. Cuticular protein gene family annotation.**
(XLSX)

**S25 Table. Juvenile hormone family annotation.**
(XLSX)

**S26 Table. Insulin signaling pathway annotation.**
(XLSX)

**S27 Table. Antioxidant family annotation.**
(XLSX)

**S28 Table. Vitamin metabolism pathway annotation.**
(XLSX)

**S29 Table.** *Phlebotomus papatasi* **population sequencing median coverage depth.**
(XLSX)

**S30 Table.** *Phlebotomus papatasi* **population variant summary statistics.**
(XLSX)

**S31 Table.** *Phlebotomus papatasi* $F_{ST}$–**Tajima's D overlap (including 10kb upstream and downstream).**
(XLSX)

**S32 Table.** *Lutzomyia longipalpis* **population sequencing median coverage depth.**
(XLSX)

**S33 Table. Parameter values of male copulatory songs from** *Lutzomyia longipalpis* **from Araci and Olindina.**
(DOCX)

**S34 Table.** *Lutzomyia longipalpis* **genes within differentiation islands.**
(XLSX)

**S1 Fig. Conflicting phylogenetic signals.** Analysis of the gene phylogenies of individual orthologous groups identified three major topologies with sand fly-mosquito (41%), sand fly-

fly (37%), or mosquito-fly (22%) sister clades. Comparisons of average branch lengths for each topology suggest that, although substitution rates in flies are always higher, orthologs that support the sand fly-mosquito topology show the lowest substitution rates in flies and the smallest differences in substitution rates among the fly, sand fly, and mosquito clades. In contrast, the sand fly-fly and mosquito-fly topologies show much higher substitution rates in flies and much greater differences in substitution rates among the three clades.
(TIF)

**S2 Fig. Clustering of sand fly galectin protein sequences.** Condensed Neighbor-Joining tree depicting clustering among galectin protein sequences of sand flies (*Ph. papatasi* and *Lu. longipalpis*; open and filled squares, respectively), mosquitoes (*Ae. aegypti* and *An. gambiae*; open and filled circles, respectively), fly (*D. melanogaster*; filled triangle), eastern oyster (*C. virginica*; upside-down open triangle), and freshwater snail (*B. glabrata*; upside-down filled triangle). Branches encompassing shared orthologs are highlighted by blue shades. Sand fly specific clusters and genes are highlighted by orange shades. The evolutionary distances were computed using the p-distance method and are in the units of the number of amino acid differences per site. One thousand bootstrap replicates were performed, and only branches displaying at least 50% confidence are shown.
(TIF)

**S3 Fig. Condensed Neighbor-Joining tree depicting clustering among n-acetylhexosaminidase protein sequences of sand flies (*Ph. papatasi* and *Lu. longipalpis*; open and filled squares, respectively), mosquitoes (*Ae. aegypti* and *An. gambiae*; open and filled circles, respectively), fly (*D. melanogaster*; filled triangle), and beetle (*T. castaneum*; filled diamond).** Branches encompassing sequences belonging to group I-IV n-acetylhexosaminidases are highlighted by a blue shade. The sand fly specific cluster is highlighted by an orange shade. The evolutionary distances were computed using the p-distance method and are in the units of the number of amino acid differences per site. One thousand bootstrap replicates were performed, and only branches displaying at least 50% confidence are shown.
(TIF)

**S4 Fig. Condensed Neighbor-Joining tree depicting clustering among chitin deacetylase catalytic domain sequences of sand flies (*Ph. papatasi* and *Lu. longipalpis*; open and filled squares, respectively), mosquitoes (*Ae. aegypti* and *An. gambiae*; open and filled circles, respectively), fly (*D. melanogaster*; filled triangle), and beetle (*T. castaneum*; filled diamond).** Branches encompassing sequences belonging to group 1–5 and 9 CDA are highlighted by blue shades. The evolutionary distances were computed using the p-distance method and are in the units of the number of amino acid differences per site. One thousand bootstrap replicates were performed, and only branches displaying at least 50% confidence are shown.
(TIF)

**S5 Fig. Condensed Maximum likelihood tree depicting peritrophin CBD domain similarities among the sand flies *Ph. papatasi* and *Lu. longipalpis* and the red flour beetle *T. castaneum*.** Open squares, filled squares, and filled diamonds represent *Ph. papatasi*, *Lu. longipalpis*, and *T. castaneum* domains, respectively. Branches exclusive to *T. castaneum* were color-coded in magenta; those specific to sand flies were highlighted in blue. The branch encompassing the CBD-like domain "CBDput" is highlighted in green. The branches shared by sand flies and RFB CBD domains are color-coded in orange. Maximum likelihood tree was constructed using the Whelan and Goldman (WAG) model with Gamma distributed among Invariant sites (G+I), as suggested by the Model test function of the Mega6 software. One

thousand bootstrap replicates were performed, and only branches displaying at least 50% confidence are shown.
(TIF)

**S6 Fig. Comparison of predicted aquaporins from other flies.** Neighbor-joining tree was produced using MEGA6 using Dayhoff Model and pairwise matching; branch values indicate support following 3000 bootstraps; values below 50% are omitted.
(TIF)

**S7 Fig. Molecular phylogenetic analysis of vertebrate and invertebrate photolyases containing *Lu. longipalpis* and *Ph. papatasi* gene models.** The different photoyases are displayed on the right. The evolutionary history was inferred by using the Maximum Likelihood method based on the Jones-Taylor-Thorton + four gamma categories with 1000 bootstrap replicates (showing only above 65). Sequences with squares are vertebrate cryptochromes (black—cry-4, white—cry-1, cry-2, and cry-3); sequences with black traingles represent (6–4) insect photolyases; sequences with inverted black triangles are reprenting all insect photolyase repir proteins; and sequences with a dot symbol show insect cryptochromes (black–cry-1, white–cry-2). Dashed arrows point to *Ph. papatasi* photolyase sequences and straight arrows to *Lu. longipalpois* photolyase sequences.
(TIF)

**S8 Fig. Molecular phylogenetic analysis of *Lu. longipalpis*, *Ph. papatasi* and *D. melanogaster* TRP channel sequences.** The different TRP subfamilies are displayed on the right. The evolutionary history was inferred by using the Maximum Likelihood method based on the Whelan and Goldman +Freq. model with 1000 bootstrap replicates.
(TIF)

**S9 Fig. Molecular phylogenetic analysis of *Lu. longipalpis*, *Ph. papatasi* and *D. melanogaster* PPK sequences.** The different PPK subfamilies are displayed on the right. The evolutionary history was inferred by using the Maximum Likelihood method based on the Whelan and Goldman +Freq. model with 1000 bootstrap replicates.
(TIF)

**S10 Fig. Maximum likelihood phylogeny of the manually curated CYPs in sand flies *Lu. longipalpis* (name shown in blue) and *Ph. papatasi* (names shown in red).** CYPome of the mosquito *An. gambiae* (names shown in orange) was used as reference, while the tree was rooted using the human CYP51A1 as an outgroup. All four insect CYP clans are well-supported with bootstrap values >95%. The leafs representing the CYP9J/9L, CYP6AG and CYP6AK expansions in *Lu. longipalpis* and *Ph. papatasi* are highlighted with cyan, grey and green, respectively. Branches for each of the four different insect CYP clans are colored differently; Mito clan—cyan, CYP2 clan—gold, CYP3 clan—green, CYP4 clan—orange.
(TIF)

**S11 Fig. *Phlebotomus papatasi* population structure.** Inferred population structure of *Ph. papatasi* individuals collected from Afghanistan (PPAFG; green), North Sinai—Egypt (PPNS; purple), and Tunisia (PPTUN; orange). (A) Phylogenetic Analysis. Rooted neighbor joining (NJ) radial tree generated with the Adegenet and ape packages of R. We included both *Ph. bergeroti* (PBRG; black) and *Ph. duboscqi* (PDMA; gray), and used *Ph. duboscqi* to root the trees. Bootstrap values represent the percentage of 1,000 replicates. (B) Principle component analysis (PCA). Individuals are plotted according to their coordinates on the first two principal components (PC1 and PC2). (C) Admixture analysis. Ancestry proportions for Admixture models from $K = 2$ to $K = 7$ ancestral populations. Each individual is represented by a thin vertical

line, partitioned into *K* coloured segments representing the individual's estimated membership fractions to the *K* clusters. These data are the average of the major *q*-matrix clusters derived by CLUMPAK analysis.
(TIF)

**S12 Fig. Admixture cross validation error.** Violin plot of the cross-validation error for each of 30 replicates for each *K* value. (A) *Phlebotomus papatasi* populations. (B) *Lutzomyia longipalpis* populations.
(TIF)

**S13 Fig. Male copulatory courtship songs from Araci and Olinda.** (A) Approximate distance of Araci and Olinda from Jacobina (B). Male copulatory courtship song tracings of *Lutzomyia longialpis* males collected from Araci and Olindina. The figure shows ~1 s of song in each case. Main map source: World Imagery (Source: Esri, Maxar, Earthstar Geographics, and the GIS User Community; http://goto.arcgisonline.com/maps/World_Imagery). Inset map source: World Dark Gray Canvas Base (Esri, HERE, Garmin, OpenStreetMap contributors, and the GIS user community; http://goto.arcgisonline.com/maps/Canvas/World_Dark_Gray_Base).
(TIF)

**S14 Fig. Distribution plots of the pairwise $F_{ST}$ between the different populations of *Lutzomyia longipalpis*.** Weighted $F_{ST}$ values for 1kb non-overlapping windows were calculated across the genome for each population comparison.
(TIF)

**S15 Fig. Manhattan plots of the pairwise $F_{ST}$ between the different populations of *Lutzomyia longipalpis*.** The red horizontal lines indicate the upper 0.05% of $F_{ST}$ distribution over the entire genome.
(TIF)

**S16 Fig. Manhattan plot of Tajimas'D for each population of *Lutzomyia longipalpis*.** The red and blue horizontal lines indicate the upper and lower 0.05% of Tajima's D distribution, respectively.
(TIF)

**S17 Fig. Genomic regions with high (outlier) Tajimas'D for different populations of *Lutzomyia longipalpis*.** (A) The Venn diagram summarizes the numbers of 1kb genomic windows with Tajimas'D values in the upper 2.5% of the different populations. (B) The Venn diagram summarizes the numbers of 1kb genomic windows with Tajimas'D values in the lower 2.5% of the different populations.
(TIF)

**S1 Data. *Phlebotomus papatasi* CYPome Fasta File.** Open with a text editor.
(FASTA)

**S2 Data. *Lutzomyia longipalpis* CYPome Fasta File.** Open with a text editor.
(FASTA)

## Acknowledgments

Two of our collaborators, Dr Alexandre A. Peixoto (Feb. 2013) and Dr. Hector M. Diaz Albiter (Feb. 2021) were influential in this project but died before the work could be completed. We dedicate this manuscript to them in recognition of contributions to both this work and the vector disease community as a whole. In addition, gratitude goes to Sandra Clifton who was

instrumental in the work but retired before completion and to Daniel Lawson who was instrumental in the initial stages of the project while a member of the VectorBase staff. The authors thank the sequencing team members of the Washington University Genome Center and Baylor College of Medicine Human Genome Sequencing Center for their efforts in generating the raw sequence data.

## Author Contributions

**Conceptualization:** Rafaela V. Bruno, Fernando A. Genta, Shaden Kamhawi, Jesus Valenzuela, Stephen Richards, Rod J. Dillon, Mary Ann McDowell.

**Data curation:** Frédéric Labbé, Maha Abdeladhim, Jenica Abrudan, Alejandra Saori Araki, Ricardo N. Araujo, Peter Arensburger, Joshua B. Benoit, Rafaela V. Bruno, Gustavo Bueno da Silva Rivas, Vinicius Carvalho de Abreu, Jason Charamis, Iliano V. Coutinho-Abreu, Samara G. da Costa-Latgé, Alistair Darby, Viv M. Dillon, Scott J. Emrich, Daniela Fernandez-Medina, Nelder Figueiredo Gontijo, Catherine M. Flanley, Derek Gatherer, Fernando A. Genta, Gloria I. Giraldo-Calderón, Bruno Gomes, Eric Roberto Guimaraes Rocha Aguiar, Omar Hamarsheh, Mallory Hawksworth, Jacob M. Hendershot, Paul V. Hickner, Jean-Luc Imler, Panagiotis Ioannidis, Emily C. Jennings, Charikleia Karageorgiou, Ryan C. Kennedy, José M. Latorre-Estivalis, Antonio Carlos A. Meireles-Filho, Michael J. Montague, Ronald J. Nowling, Fabiano Oliveira, João Ortigão-Farias, Marcio G. Pavan, Marcos Horacio Pereira, Andre Nobrega Pitaluga, Roenick Proveti Olmo, Marcelo Ramalho-Ortigao, José M. C. Ribeiro, Andrew J. Rosendale, Mauricio R. V. Sant'Anna, Steven E. Scherer, Caroline da Silva Moraes, João Silveira Moledo Gesto, Nataly Araujo Souza, Samuel Tadros, Rayane Teles-de-Freitas, Erich L. Telleria, Chad Tomlinson, João Trindade Marques, Zhijian Tu, Maria F. Unger, Flávia V. Ferreira, Karla P. V. de Oliveira, Felipe M. Vigoder, Lihui Wang, Gareth D. Weedall, Stephen Richards, Robert M. Waterhouse.

**Formal analysis:** Frédéric Labbé, Maha Abdeladhim, Alejandra Saori Araki, Ricardo N. Araujo, Peter Arensburger, Joshua B. Benoit, Rafaela V. Bruno, Gustavo Bueno da Silva Rivas, Vinicius Carvalho de Abreu, Jason Charamis, Iliano V. Coutinho-Abreu, Samara G. da Costa-Latgé, Alistair Darby, Viv M. Dillon, Scott J. Emrich, Daniela Fernandez-Medina, Nelder Figueiredo Gontijo, Catherine M. Flanley, Derek Gatherer, Fernando A. Genta, Sandra Gesing, Gloria I. Giraldo-Calderón, Bruno Gomes, Eric Roberto Guimaraes Rocha Aguiar, Omar Hamarsheh, Mallory Hawksworth, Jacob M. Hendershot, Paul V. Hickner, Jean-Luc Imler, Panagiotis Ioannidis, Emily C. Jennings, Shaden Kamhawi, Charikleia Karageorgiou, Ryan C. Kennedy, José M. Latorre-Estivalis, Petros Ligoxygakis, Antonio Carlos A. Meireles-Filho, Patrick Minx, Michael J. Montague, Ronald J. Nowling, Fabiano Oliveira, João Ortigão-Farias, Marcio G. Pavan, Marcos Horacio Pereira, Andre Nobrega Pitaluga, Roenick Proveti Olmo, Marcelo Ramalho-Ortigao, José M. C. Ribeiro, Andrew J. Rosendale, Mauricio R. V. Sant'Anna, Steven E. Scherer, Caroline da Silva Moraes, João Silveira Moledo Gesto, Nataly Araujo Souza, Samuel Tadros, Rayane Teles-de-Freitas, Erich L. Telleria, Chad Tomlinson, João Trindade Marques, Zhijian Tu, Maria F. Unger, Flávia V. Ferreira, Karla P. V. de Oliveira, Felipe M. Vigoder, John Vontas, Lihui Wang, Gareth D. Weedall, Stephen Richards, Robert M. Waterhouse, Rod J. Dillon.

**Funding acquisition:** Rod J. Dillon.

**Investigation:** Frédéric Labbé, Peter Arensburger, Joshua B. Benoit, Reginaldo Pecanha Brazil, Rafaela V. Bruno, Scott J. Emrich, Sandra Gesing, Gloria I. Giraldo-Calderón, Omar Hamarsheh, Paul V. Hickner, Ryan C. Kennedy, Patrick Minx, Michael J. Montague,

Ronald J. Nowling, João Ortigão-Farias, Douglas A. Shoue, Chad Tomlinson, Gareth D. Weedall, Stephen Richards, Robert M. Waterhouse.

**Methodology:** Alejandra Saori Araki, Sandra Gesing, Patrick Minx, Ronald J. Nowling, Chad Tomlinson.

**Project administration:** Stephen Richards, Wesley C. Warren, Rod J. Dillon, Mary Ann McDowell.

**Resources:** Reginaldo Pecanha Brazil, Andreas Krueger, Jose Carlos Miranda, Nágila F. C. Secundino, Elyes Zhioua.

**Supervision:** Joshua B. Benoit, Rafaela V. Bruno, Alistair Darby, Scott J. Emrich, Derek Gatherer, Fernando A. Genta, Gloria I. Giraldo-Calderón, Panagiotis Ioannidis, Petros Ligoxygakis, Antonio Carlos A. Meireles-Filho, João Ortigão-Farias, Zainulabueddin Syed, Yara M. Traub-Csekö, Jesus Valenzuela, John Vontas, Wesley C. Warren, Mary Ann McDowell.

**Validation:** Maha Abdeladhim, Alejandra Saori Araki, Scott J. Emrich, Derek Gatherer, James G. C. Hamilton, Mary Ann McDowell.

**Visualization:** Robert M. Waterhouse, Mary Ann McDowell.

**Writing – original draft:** Frédéric Labbé, Maha Abdeladhim, Alejandra Saori Araki, Ricardo N. Araujo, Peter Arensburger, Joshua B. Benoit, Rafaela V. Bruno, Gustavo Bueno da Silva Rivas, Vinicius Carvalho de Abreu, Jason Charamis, Iliano V. Coutinho-Abreu, Samara G. da Costa-Latgé, Alistair Darby, Viv M. Dillon, Scott J. Emrich, Daniela Fernandez-Medina, Nelder Figueiredo Gontijo, Derek Gatherer, Fernando A. Genta, Bruno Gomes, Eric Roberto Guimaraes Rocha Aguiar, Omar Hamarsheh, Jacob M. Hendershot, Paul V. Hickner, Jean-Luc Imler, Panagiotis Ioannidis, Emily C. Jennings, Shaden Kamhawi, Charikleia Karageorgiou, José M. Latorre-Estivalis, Petros Ligoxygakis, Antonio Carlos A. Meireles-Filho, Patrick Minx, Michael J. Montague, Ronald J. Nowling, Fabiano Oliveira, João Ortigão-Farias, Marcio G. Pavan, Marcos Horacio Pereira, Andre Nobrega Pitaluga, Roenick Proveti Olmo, Marcelo Ramalho-Ortigao, José M. C. Ribeiro, Andrew J. Rosendale, Mauricio R. V. Sant'Anna, Steven E. Scherer, Caroline da Silva Moraes, João Silveira Moledo Gesto, Nataly Araujo Souza, Zainulabueddin Syed, Rayane Teles-de-Freitas, Erich L. Telleria, Yara M. Traub-Csekö, João Trindade Marques, Zhijian Tu, Maria F. Unger, Flávia V. Ferreira, Karla P. V. de Oliveira, Felipe M. Vigoder, John Vontas, Lihui Wang, Gareth D. Weedall, Stephen Richards, Wesley C. Warren, Robert M. Waterhouse, Rod J. Dillon, Mary Ann McDowell.

**Writing – review & editing:** Frédéric Labbé, Maha Abdeladhim, Joshua B. Benoit, Rafaela V. Bruno, Gustavo Bueno da Silva Rivas, Scott J. Emrich, Derek Gatherer, Fernando A. Genta, Bruno Gomes, James G. C. Hamilton, Paul V. Hickner, Panagiotis Ioannidis, Andreas Krueger, José M. Latorre-Estivalis, Petros Ligoxygakis, Antonio Carlos A. Meireles-Filho, Patrick Minx, Ronald J. Nowling, Fabiano Oliveira, Marcelo Ramalho-Ortigao, José M. C. Ribeiro, Douglas A. Shoue, Zainulabueddin Syed, Erich L. Telleria, Chad Tomlinson, Gareth D. Weedall, Stephen Richards, Wesley C. Warren, Robert M. Waterhouse, Rod J. Dillon, Mary Ann McDowell.

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
