## [Decision Letter · Decision Letter 0]

11 Nov 2022

Dear Dr. McDowell,

Thank you very much for submitting your manuscript "Genomic Analysis of Two Phlebotomine Sand Fly Vectors of Leishmania from the New and Old World" for consideration at PLOS Neglected Tropical Diseases. As with all papers reviewed by the journal, your manuscript was reviewed by members of the editorial board and by several independent reviewers. The reviewers appreciated the attention to an important topic. Based on the reviews, we are likely to accept this manuscript for publication, providing that you modify the manuscript according to the review recommendations. 

The reviewers have completed their evaluations of the manuscript and based on those we have come to a decision of minor revision. Reviewer 2 has made a decision of "Minor Revision" and raises some issues that should be addressed by the authorship. Both reviewers felt the paper was an important contribution to the sandfly and more broadly vector genomics community. We look forward to the submission of the revised version of your manuscript.

Sincerely,

Geoffrey M. Attardo

Academic Editor

Alvaro Acosta-Serrano

Section Editor

Thank you for the submission of the manuscript "Genomic Analysis of Two Phlebotomine Sand Fly Vectors of Leishmania from the New and Old World". The reviewers have completed their evaluations of the manuscript and based on those we have come to a decision of minor revision. Reviewer 2 has made a decision of "Minor Revision" and raises some issues that should be addressed by the authorship. Both reviewers felt the paper was an important contribution to the sandfly and more broadly vector genomics community. We look forward to the submission of the revised version of your manuscript.

Reviewer's Responses to Questions

**Key Review Criteria Required for Acceptance?**

**Methods**

-Are the objectives of the study clearly articulated with a clear testable hypothesis stated?

-Is the study design appropriate to address the stated objectives?

-Is the population clearly described and appropriate for the hypothesis being tested?

-Is the sample size sufficient to ensure adequate power to address the hypothesis being tested?

-Were correct statistical analysis used to support conclusions?

-Are there concerns about ethical or regulatory requirements being met?

Reviewer #1: The methods are appropriate and described in sufficient detail to be reproducible. I see no difficulties as written.

Reviewer #2: -Line 246. It is a stretch to say that DNA was sourced from inbred insects when no effort was made to make an inbred line. Other groups at the time performed several generations of full sibling single brother/sister matings to make an inbred line. Any claim to using an inbred strain should be removed. To say the DNA was sourced from a lab strain that may have reduced heterozygosity as the colony fluctuated in colony size over time would be acceptable.

-Paragraph beginning line 315. Quite a few details on the genome assembly that would normally be included are missing here. How were contaminants removed? Cite method used. Citations for the in the in-house “PolyGraph assembler” and “PyGap” are missing. Please add or provide more details on the software. They should be made available (e.g. GitHub). Illumina data was used for gap closure..but where did this come from? Only Sanger and 454 reads are mentioned in the preceding paragraph.

-line 353. List the stages sourced for RNAseq and ages of adults.

**Results**

-Does the analysis presented match the analysis plan?

-Are the results clearly and completely presented?

-Are the figures (Tables, Images) of sufficient quality for clarity?

Reviewer #1: The Results are clearly presented, with appropriate supporting Figures.

Reviewer #2: _lines 522-528. N50s of 28kb and 85 kb and BUSCO scores of 86% are quite poor by current standards. Very fragmented genomes that are likely missing regions of the genomes. 

-orthology. 36 dipteran species sounds impressive but it is mostly a comparison of the two sandfly genomes with Drosophila and Anopheles genomes. So, not a broad sampling of Diptera. This was likely done at least 8 years ago. My suggestion would be to redo this analysis with a broader sampling of Dipteran genomes and not such a heavy reliance on Drosophila and Anopheles.

-line 564. My recommendation would be to delete this section. With the current quality of reference genomes, similar synteny analyses use long chromosome-length scaffolds. Perhaps microsynteny could be reported but I would think this would be better left for the next versions of the genomes of these species.

-TEs. With such fragmented genomes are the authors confident they can accurately report the % of the genomes that are repetitive? Long reads would definitely help with the assembly of regions with blocks of repetitive DNA. Consequently, speculation on the possible expansion of TEs in one species would seem premature. Also, it is class II, the DNA transposons, that are “copy and paste” not class I, retrotransposons.

-Circadian Rhythm Genes. Perhaps Lu. longipalpis has only one CRY gene but with such highly fragmented genomes, this is not particularly convincing. Is this supported by RNAseq data? This section could be deleted. 

-Chemosensory Receptors. Wasn’t this already published? If nothing new here this section could be deleted.

-No comments on the population structure section other than I thought it was quite interesting and suggests follow-up studies are needed in Jacobina given the data suggests there are two groups.

**Conclusions**

-Are the conclusions supported by the data presented?

-Are the limitations of analysis clearly described?

-Do the authors discuss how these data can be helpful to advance our understanding of the topic under study?

-Is public health relevance addressed?

Reviewer #1: (No Response)

Reviewer #2: OK

**Editorial and Data Presentation Modifications?**

Reviewer #1: Recommendation: acceptance without revision

I hesitate to suggest more work that might delay publication, because I believe this paper will be valuable for other members of the dipteran genome community. However, if the authors might find it helpful to include the Chironomidae in their orthology analysis. Several chironomids genomes are available in OrthoDB (e.g., Clunio). Chironomids are members of the Culicimorpha, as sister family to the Simuliidae (blackflies). Although they are not vectors, having a non-mosquito member of the Culicimorpha might help strengthen the clustering of the phlebotomines with the Culicimorphs as opposed to the advanced dipterans (or perhaps change the topology to the contrary!). However, I certainly would not make acceptance contingent on inclusion of chironomids.

Reviewer #2: -Paragraph beginning line 211. The transition from blood meal digestion to circadian rhythms did not flow well. I realize that this sets up the finding that one sandfly species may only have one CRY gene but this deserves a separate paragraph. Or this could be removed entirely (see comments on results)

**Summary and General Comments**

Reviewer #1: This paper is a significant contribution to the study, not only of the phlebotomine sandflies, but also the wider group of dipteran disease vectors. The authors are to be congratulated on their success with these very challenging genome assemblies, and the extensive analysis of the genomic data. Personal highlights for me include the treatment of transposable element families and circadian rhythm genes, and also immunity genes (my current rabbit-hole in the family that I work with). The inclusion of natural population samples is a valuable enhancement to the analysis.

Reviewer #2: It can take a long time from the completion of a genome assembly to publication but not usually a decade! The genome assemblies for the two sandfly species were completed in 2012 from a mix of Sanger and 454 reads! The DNAs were sourced from lab strains that had undergone bottlenecks but were not deliberately inbred for multiple generations such as was done for other insect genome projects at the time. Consequently, the assemblies are highly fragmented and not what would be considered reference genomes in 2022. Most insect genome papers I have seen this year use long reads such as PacBio HiFi and use HiC to obtain chromosome-length scaffolds. While the genome assemblies do not meet current standards for reference genomes, the manuscript is important because of the extensive efforts the authors have made at manual annotation and use of the assemblies for an analysis of population structures. A huge amount of valuable information has been collated in the 36 supplementary tables and 18 supplementary figures. I mostly have a few minor comments that should be addressed and one request. My request is that, if not done already, that all the manual annotations be submitted to NCBI. Too often I find insect genomes submitted to NCBI with only the annotations generated by MAKER (or similar) with the manual annotations only provided in the supplementary data of the paper.

PLOS authors have the option to publish the peer review history of their article (what does this mean?). If published, this will include your full peer review and any attached files.

Reviewer #1: No

Reviewer #2: No

Figure Files:

Data Requirements:

Reproducibility:

References

---

## [Decision Letter · Decision Letter 1]

13 Feb 2023

Dear Dr. McDowell,

We are pleased to inform you that your manuscript 'Genomic Analysis of Two Phlebotomine Sand Fly Vectors of Leishmania from the New and Old World' has been provisionally accepted for publication in PLOS Neglected Tropical Diseases.

Best regards,

Geoffrey M. Attardo

Academic Editor

Alvaro Acosta-Serrano

Section Editor

Thank you very much for submitting the revised version of your manuscript addressing the comments from the reviewers. I'm happy to inform you that the reviewers are satisfied with the revisions and the manuscript is ready for acceptance and publication!

Reviewer's Responses to Questions

**Key Review Criteria Required for Acceptance?**

**Methods**

-Are the objectives of the study clearly articulated with a clear testable hypothesis stated?

-Is the study design appropriate to address the stated objectives?

-Is the population clearly described and appropriate for the hypothesis being tested?

-Is the sample size sufficient to ensure adequate power to address the hypothesis being tested?

-Were correct statistical analysis used to support conclusions?

-Are there concerns about ethical or regulatory requirements being met?

Reviewer #1: Having been over the revised version of the manuscript, my previous recommendation of acceptance stands. I did not request many revisions, and the authors made a good effort at complying with the recommendations of reviewer #2.

This manuscript remains an important contribution to Dipteran genomics. The addition of more non-mosquito Nematoceran genomes is essential, and the enhancement made by examining natural population is significant.

Reviewer #2: (No Response)

**Results**

-Does the analysis presented match the analysis plan?

-Are the results clearly and completely presented?

-Are the figures (Tables, Images) of sufficient quality for clarity?

Reviewer #1: No additional comment to my previous review.

Reviewer #2: (No Response)

**Conclusions**

-Are the conclusions supported by the data presented?

-Are the limitations of analysis clearly described?

-Do the authors discuss how these data can be helpful to advance our understanding of the topic under study?

-Is public health relevance addressed?

Reviewer #1: No additional comment to my previous review.

Reviewer #2: (No Response)

**Editorial and Data Presentation Modifications?**

Reviewer #1: No further modifications necessary.

Reviewer #2: (No Response)

**Summary and General Comments**

Reviewer #1: This paper remains an important work, with significant impact on Dipteran, and in particular Nematoceran, genomics.

Reviewer #2: the authors have addressed my concerns

PLOS authors have the option to publish the peer review history of their article (what does this mean?). If published, this will include your full peer review and any attached files.

Reviewer #1: No

Reviewer #2: No

---

## [Editor Report · Acceptance letter]

5 Apr 2023

Dear Dr. McDowell,

We are delighted to inform you that your manuscript, "Genomic Analysis of Two Phlebotomine Sand Fly Vectors of *Leishmania* from the New and Old World," has been formally accepted for publication in PLOS Neglected Tropical Diseases.

Best regards,

Shaden Kamhawi

co-Editor-in-Chief

Paul Brindley

co-Editor-in-Chief
